# Flexible, sticky, and biodegradable wireless device for drug delivery to brain tumors

Jongha Lee[1,2,10], Hye Rim Cho[1,3,10], Gi Doo Cha[1,2,10], Hyunseon Seo[1,2,4], Seunghyun Lee[3], Chul-Kee Park[5], Jin Wook Kim[5], Shutao Qiao[6], Liu Wang[6], Dayoung Kang[1,2], Taegyu Kang[1,2], Tomotsugu Ichikawa[7], Jonghoon Kim[1,2], Hakyong Lee[1,2], Woongchan Lee[1,2], Sanghoek Kim[8], Soon-Tae Lee[9], Nanshu Lu[6], Taeghwan Hyeon[1,2], Seung Hong Choi[1,3]* & Dae-Hyeong Kim[1,2]*

Implantation of biodegradable wafers near the brain surgery site to deliver anti-cancer agents which target residual tumor cells by bypassing the blood-brain barrier has been a promising method for brain tumor treatment. However, further improvement in the prognosis is still necessary. We herein present novel materials and device technologies for drug delivery to brain tumors, i.e., a flexible, sticky, and biodegradable drug-loaded patch integrated with wireless electronics for controlled intracranial drug delivery through mild-thermic actuation. The flexible and bifacially-designed sticky/hydrophobic device allows conformal adhesion on the brain surgery site and provides spatially-controlled and temporarily-extended drug delivery to brain tumors while minimizing unintended drug leakage to the cerebrospinal fluid. Biodegradation of the entire device minimizes potential neurological side-effects. Application of the device to the mouse model confirms tumor volume suppression and improved survival rate. Demonstration in a large animal model (canine model) exhibited its potential for human application.

[1] Center for Nanoparticle Research, Institute for Basic Science (IBS), Seoul 08826, Republic of Korea. [2] School of Chemical and Biological Engineering, Institute of Chemical Processes, Seoul National University, Seoul 08826, Republic of Korea. [3] Department of Radiology, Seoul National University College of Medicine, Seoul 03080, Republic of Korea. [4] Center for Biomaterials, Korea Institute of Science and Technology, Seoul 02792, Republic of Korea. [5] Department of Neurosurgery, Seoul National University College of Medicine, Seoul 03080, Republic of Korea. [6] Center for Mechanics of Solids, Structures and Materials, Department of Aerospace Engineering and Engineering Mechanics, University of Texas at Austin, Austin, TX 78712, USA. [7] Department of Neurological surgery, Okayama University Graduate School of Medicine, Dentistry, and Pharmaceutical Sciences, Okayama 700-8558, Japan. [8] Department of Electronics and Radio Engineering, Kyung Hee University, Gyeonggi 17194, Republic of Korea. [9] Department of Neurology, Seoul National University College of Medicine, Seoul 03080, Republic of Korea. [10] These authors contributed equally: Jongha Lee, Hye Rim Cho, Gi Doo Cha. *email: verocay@snuh.org; dkim98@snu.ac.kr

Treatment of cancers in organs with special blood barriers such as brain, peritoneum, and oculus has always been challenging. Especially, treatment of malignant brain tumors (e.g., glioblastoma; GBM) is extremely difficult[1], because tumor cells survive through surgical resection and radiation therapy[2,3], causing tumor recurrence. Conventional chemotherapies through the intravenous delivery are oftentimes unsuccessful since the blood–brain barrier blocks delivery of drugs to brain tumors[4]. Several researches have improved controlled and targeted drug delivery to brain tumors[5–7]. For example, the biodegradable polymeric wafer[8,9] (Gliadel wafer, Arbor Pharmaceuticals, USA) implanted near the brain surgery site locally delivers drugs to remaining brain tumors and exhibits meaningful improvement. However, further advances in the treatment efficacy are still necessary.

The requirements of the device for the controlled intracranial drug delivery to brain tumors are as follows. The penetration[10] of the released drugs into brain tissues should be high for treatment of infiltrated tumor cells. Local drug delivery to brain tumors without unwanted release to cerebrospinal fluid (CSF) is important[11]. Mechanical mismatch of rigid implantable devices from brain tissues may cause neurological disorders, and thereby soft electronic devices are preferred for intracranial implants[12–14]. Devices with the capability of complete bioresorption can be a good solution to prevent side effects of chronic implants[15–18]. Long-term sustained drug delivery is helpful for cancer treatment[19,20]. Toward accomplishment of these challenging goals altogether, a novel soft biodegradable electronic device that can actuate drug diffusion wirelessly and disappears after a desired period of time[17,21–23] in the brain[15,16] is needed.

Here, we report materials and device technologies for a flexible, sticky, and biodegradable wireless electronic device integrated with a bifacially designed polymer drug reservoir, which is called as a bioresorbable electronic patch (BEP). The BEP, together with an associated mild-thermic actuation protocol, provides long drug diffusion length and drug delivery duration. The flexibility of the oxidized starch (OST)-based patch and its hydrophilic/hydrophobic bifacial design allow conformal adhesion to the target brain tissue[24,25] and enable local and sustained drug delivery, while reducing unintended drug release to CSF. Fully bioresorbable and soft nature of the BEP minimizes potential neurological side effects of rigid intracranial implants[26]. Wireless mild-thermic actuation by the bioresorbable heater with the alternating magnetic field enhances the penetration depth of delivered drugs. The synergetic effect on brain tumor treatment by integration of all these material and device components is confirmed in mouse subcutaneous and canine brain GBM models in vivo.

## Results

**Overview of materials and devices.** The integrated device, BEP, has a bifacial structure that is composed of a hydrophilic drug-loaded OST film and a hydrophobic poly(lactic acid) (PLA) encapsulation film (Fig. 1a). Magnesium-based ultrathin electronic devices, which work as a wireless heater for mild-thermic drug delivery actuation and a wireless temperature sensor for controlled mild-thermic actuation, are embedded in these thin films (Fig. 1b, left). All material elements of the BEP are bioresorbable, and all biodegradation products are materials existing in the human body[27–30] (Fig. 1b, right). The detailed device fabrication process is described in Supplementary Fig. 1. The BEP was packaged (Fig. 1c), sterilized before implantation (Fig. 1d), and applied to animal models for drug delivery to GBM (Fig. 1e, f). Figure 1e shows the application process of the BEP in a canine GBM model during the craniotomy (Fig. 1e, left) and lamination

of the BEP on the surgical site (Fig. 1e, right). The flexible and sticky design of the BEP facilitates its conformal adhesion to the curved brain cavity surface (Fig. 1f). The strong adhesion of the hydrophilic OST bottom drug reservoir is due to the imine conjugation (Fig. 1a, right). These conformal and strong adhesion improves the efficiency of the drug delivery. Meanwhile, the hydrophobic PLA top encapsulation reduces undesirable drug delivery to CSF. The imine conjugation also helps long-term sustained drug delivery. Then, an alternating radio frequency (RF) magnetic field (220 kHz, 360A; Easyheat, Ambrell, USA) is applied to actuate the heater in the BEP wirelessly, which releases drugs from the reservoir, accelerates the intercellular drug diffusion, and enhances the drug penetration depth (Fig. 1f). The alternating RF magnetic field allows long-distance wireless energy transfer through tissues[31], which activates the mild-thermic actuation to promote drug diffusion to microscopic residual tumors invaded in normal brain tissues[32].

**Local drug delivery and biocompatibility.** We first designed materials for the BEP. It should conformally adhere to the curved brain surface for local drug delivery, for facile heat transfer during mild-thermic actuation, and for prolonged drug delivery duration by minimizing unwanted drug leakage to CSF. This could be achieved by using OST which is synthesized by oxidization of starch (Supplementary Fig. 2). It provides strong imine conjugation to both brain tissues and drug molecules (e.g., doxorubicin; DOX) (Fig. 2a, green), and thus enables good adhesion and sustained drug release. The flexible drug-loaded patch was made by mixing OST with DOX in PBS and dried under the humid condition (Supplementary Fig. 3). DOX was used as a major antitumor agent in this study because of its outstanding therapeutic effect and easy visualization using fluorescence. The maximum DOX loading amount that we have observed was 6.831 mg per one BEP since the DOX is highly miscible to the OST. The total amount of DOX can be increased further by using multiple BEPs. Further optimization to increase the drug loading amount should be done in the future. Temozolomide[33] can also be used as an alternative drug (Supplementary Fig. 3d). The BEP can also deliver multiple anticancer agents (DOX and TMZ) for the combination therapy (Supplementary Fig. 3e). Although we loaded two kinds of drugs in the BEP, the integrity of the device was well maintained.

Since the adhesion force between the OST film and the brain tissue is stronger than the mechanical strength of the brain tissue[34,35], the brain tissue is mechanically torn before the detachment of the film from the brain surface. Therefore, the effect of oxidization of starch on the adhesion strength was indirectly tested on the bovine muscle instead of the brain tissue, since the bovine muscle has higher mechanical strength than the brain tissue. The shear adhesion test (Fig. 2b, inset) of OST exhibited its strong adhesion to the muscle tissues (Fig. 1a, right) than that of unmodified starch film (non-OST case, 0% in Fig. 2b) due to imine conjugation (more oxidized units). The softness of the BEP is optimized for its conformal adhesion to the convoluted brain surface by changing the ratio of OST to glycerol (Supplementary Fig. 4). The strong adhesion and softness enable the conformal contact of the BEP at the tissue surface (Fig. 2c), which allows local drug delivery (Fig. 2c, inset, white circle). The diffusion length by the natural diffusion (Fig. 2c) is small. However, it can be increased by additional mild-thermic actuations.

The drug release from the patch at 37 °C is analyzed for 4 weeks in the phosphate-buffered saline (PBS; Fig. 2d) solution. Compared with unmodified starch, OST shows more sustained drug release, which prolongs the drug delivery duration. The

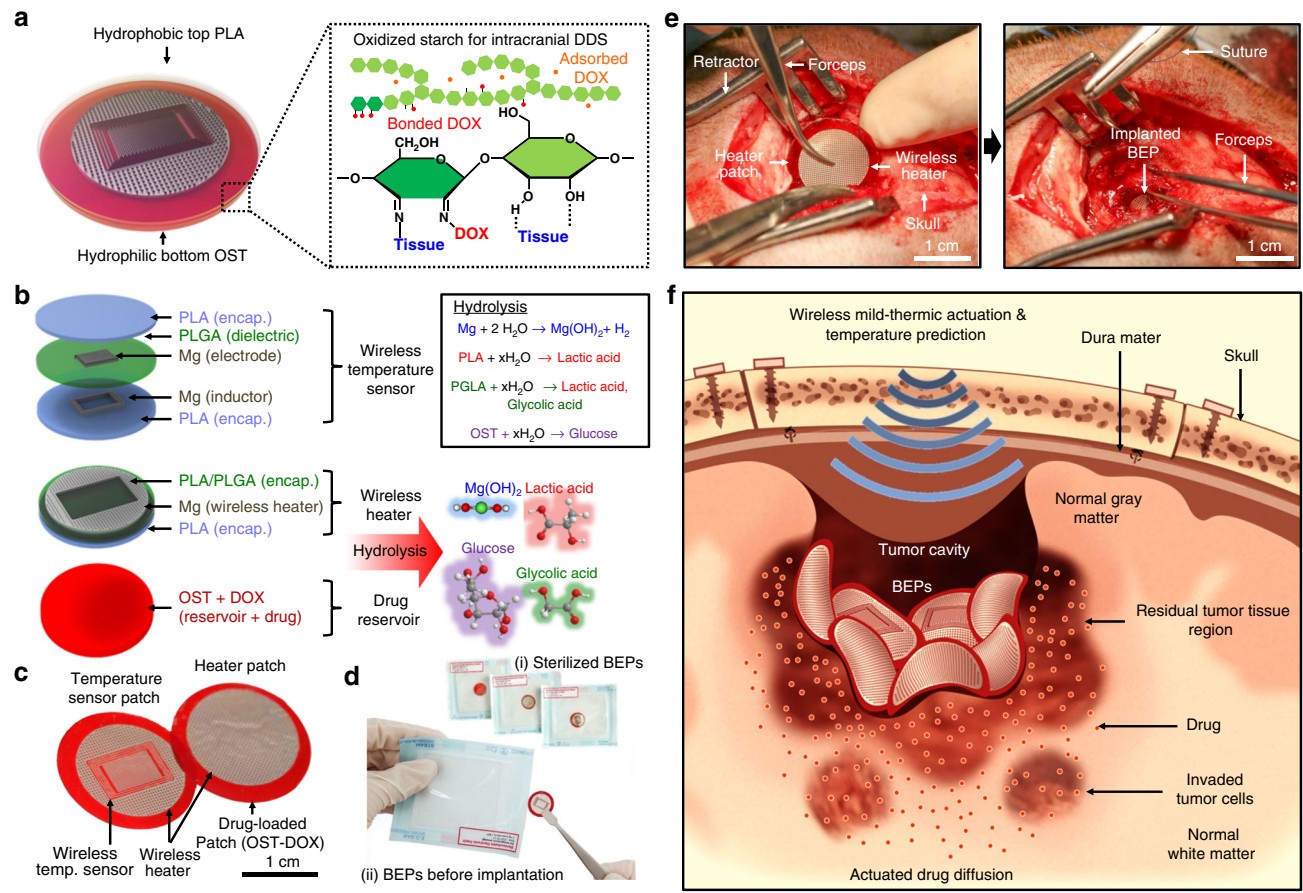

**Fig. 1** Materials, device design, and wireless actuation of the bioresorbable electronic patch (BEP). **a** Schematic illustration of the BEP (left) and the molecular structure of drug-containing oxidized starch (OST) (right). **b** Schematic illustration of the BEP and its constituent materials (left), and their biodegradation into hydrolyzed products (right). **c** Image of the BEP, which includes a bioresorbable wireless heater and a temperature sensor on an oxidized starch (OST) patch containing doxorubicin (DOX). **d** Image of the sterilized BEP before implantation. **e** Images of brain craniotomy in the canine model before (left) and after (right) BEP implantation. **f** Schematic illustration of localized and penetrative drug delivery to deep GBM tissues by the BEP with wireless mild-thermic actuation

burst drug release until day 1 (80% of DOX release) in the non-OST case (control) corresponds to release of drugs physically trapped in the polymer chain (Fig. 2a, yellow circle). The suppressed burst release at the early stage (25% DOX until day 1) and the sustained release until later periods (~50% until 4 weeks) can be achieved in OST due to chemical conjugation (i.e., hydrolysis) of drug molecules with polymer chains (Fig. 2a, red circle)[36].

When hydrophobic PLA is coated both on the top and bottom side of the OST substrate, it suppresses the drug release to the PBS solution dramatically (Fig. 2d, red). The selective coating of PLA only on the top surface of the device where the device is exposed to CSF can decrease the unintended drug leakage to other regions (Supplementary Figs. 5, 6; in vitro and in vivo, respectively), although perfect prevention of the drug diffusion to the CSF cannot be achieved due to biodegradation. Meanwhile, the bottom hydrophilic OST substrate enhances adhesion of the BEP to the brain surface and decrease the unintended drug diffusion to CSF (Fig. 2e). The DOX concentration in canine CSF in 1 week after implantation was measured to be 11.9 ng/mL at 3.15 min by high-performance liquid chromatography (HPLC) (Fig. 2f; inset shows HPLC data of DOX standard solutions). This shows minimal DOX leakage to CSF, which is important to prevent drug waste and potential side effects[37].

Fabricating the implant with materials that hydrolyze into components of human body (Fig. 1b)[24] makes the retrieval

surgery unecessary[21,38,39] and reduces risks of potential side effects of chronic neural implants[15,16]. Each component of the BEP is converted into biocompatible metabolites. For example, it takes 2 weeks for degradation of the wireless heater (Supplementary Fig. 7a, in vitro degradation; and Supplementary Fig. 7b, in vivo degradation), regardless of the mild-thermic condition (Supplementary Fig. 7c). The biodegradability of the overall device (BEP) was tested in vivo (Fig. 2g). The BEP implanted in canine brain dissolves within 10 weeks without any debris and clinical side effects (Fig. 2g). This 10-week-period corresponds to the entire duration of drug release in vivo.

In order to examine biocompatibility, the BEPs were implanted on the surface of the surgical cavity made by the brain surgery in BALB/c nude mice (Supplementary Fig. 8). The distribution of astrocytes and microglia near the surgical cavity was observed in both the sham (Fig. 2h, i, red) and BEP implantation (Fig. 2h, i, blue) group at various time points (1 day, 2 weeks, 4 weeks, and 6 weeks). No significant increase of the migrated astrocytes and microglia was observed after 2 weeks, and the differences of the migrated astrocytes and microglia between the sham group and the BEP group were not significant overall time periods. The results suggest that the BEP did not induce the significant immune response.

The implantation of the BEP to animals and its mild-thermic actuation in vivo did not affect mouse brain functions. We evaluated whether the BEP can affect the behavior of the mice.

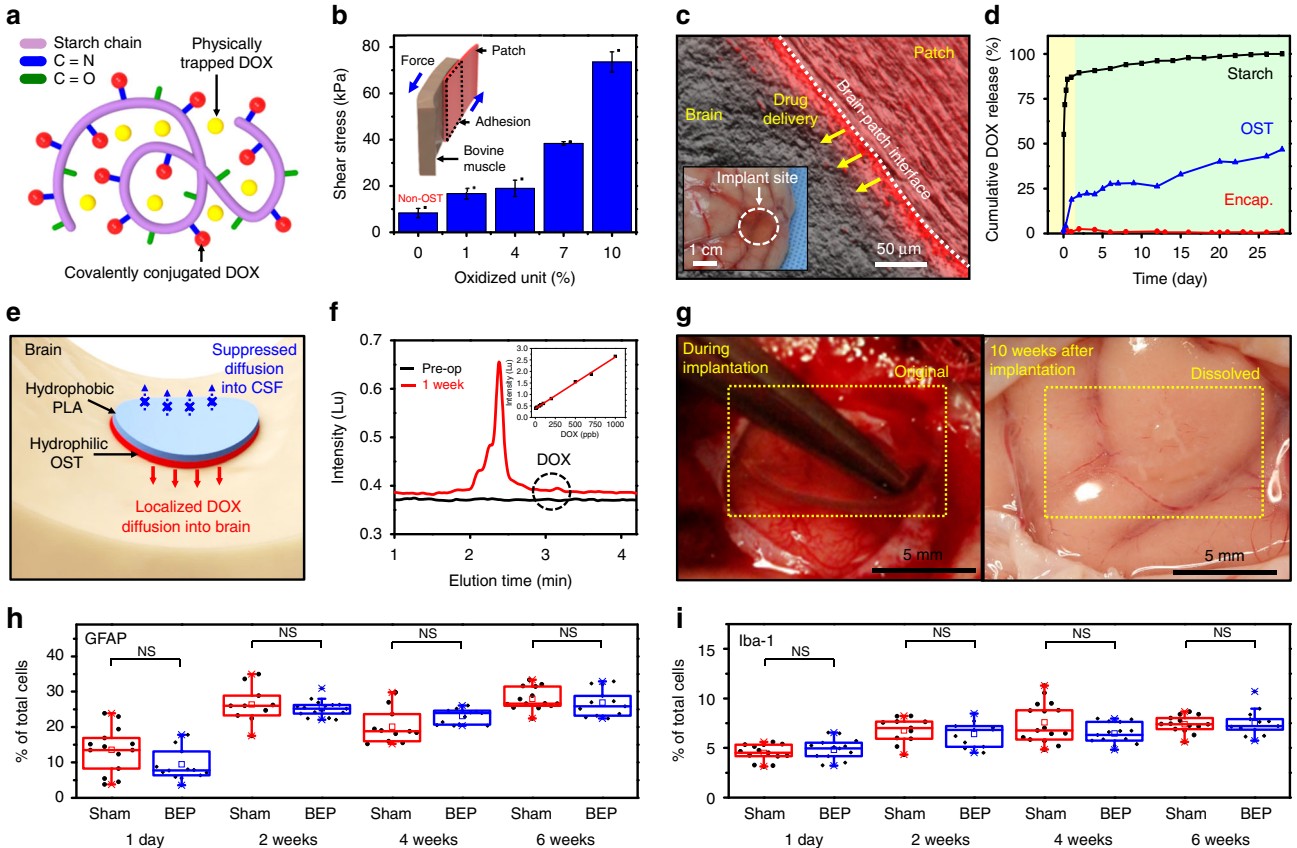

**Fig. 2** Materials, device characterization, and biocompatibiltiy. **a** Illustration of the trapped and conjugated DOX to the OST polymer chain. **b** Shear stress for detachment of the OST film with the indicated level of oxidized units from the bovine muscle tissue. Inset shows a schematic illustration of the shear stress measurement. Each experiment was repeated at least three times and error bars represent the standard error of the mean value. **c** Optical microscope image (gray) overlapped by the fluorescence microscope image (red) at the brain-BEP interface. Image of a canine brain after diffusion of DOX from the BEP (inset). **d** Cumulative release of DOX from unmodified starch (black), OST (blue), and PLA-encapsulated OST (red) in 37 °C phosphate-buffered saline (PBS) solution for 4 weeks. **e** Illustration of the flexible bifacial patch conformally adhered on the brain cavity surface. **f** Measurement of DOX concentration (dotted circle) in cerebrospinal fluid (CSF) after 1 week from implantation by HPLC. Inset shows HPLC measurement of standard DOX solutions. **g** Optical camera image during the intracranial BEP implantation surgery (left) and at 10 weeks after implantation (right). Quantification of the immunohistochemistry using BALB/c nude mice at different time points for the sham (red) and BEP implantation (blue) group (n = 7–8 for each group and time): **h** for GFAP and **i** for Iba-1. (NS; p > 0.05 by paired t-test). Line: median box: 25th–75th percentiles, Whisker: min to max, *p < 0.05 by Man–Whitney U-test with Bonferroni correction

The behavior was evaluated by the rotarod test (Supplementary Fig. 9a, b). The intracranial electroencephalogram was also monitored and analyzed (Supplementary Fig. 9c). According to the mouse tests, no neurological deficits and abnormal behaviors were observed.

The biocompatibility of the BEP was also tested in a large animal (canine model). The BEP contains MRI contrast agents, ferrimagnetic iron oxide nanocubes[40] (Supplementary Fig. 10), and thus its bioresorption and shape change in brain can be monitored by MRI in vivo (Supplementary Fig. 11). The MR images at different time points showed that the conformal contact of the BEP to the brain surface was maintained during its degradation. Compared with its initial state, the patch volume decreased significantly after 9 weeks due to its bioresorption (Supplementary Fig. 12a). Also, the intracranial implantation of the BEP to canine brain did not exhibit any unexpected side effects including brain swelling (Supplementary Fig. 11). The brain tissue reaction to the BEP was evaluated by immunochemical staining using hematoxylin and eosin (H&E) and macrophage antibody in 1 week and 10 weeks after implantation. Any significant inflammatory responses or physiological complications were not observed (Supplementary Fig. 12b). The longer-term studies to observe the effect of hydrolyzed materials in vivo are needed in the future.

**Wireless mild-thermic actuation for accelerated drug delivery.** The acceleration of drug release by wireless mild-thermic actuation (Fig. 3a) was characterized. The RF magnetic field applied by an external coil triggers eddy current and joule heating in the heater of the implanted BEP (Fig. 3b), resulting in the increased temperature of the BEP and surrounding brain tissues. Key parameters of the heater design are the diameter and thickness of the round-shaped heater while the transmission coil current and coil-to-heater distance can be varied to optimize the heat generation. The hole array in the wireless heater is helpful for facile fabrication (transfer printing) of the device but slightly affects the heat generation (Supplementary Fig. 13a, b). The temperature change under the various coil-to-heater distances in the heater (Fig. 3c) of different diameters and under the various coil-currents in 12 mm diameter heater (Fig. 3c, inset) were measured using our instrument to set the temperature change. Under different conditions, the calibration curve that shows the temperature increase as a function of the magnetic field (Supplementary Fig. 13c) and/or the total eddy current (Supplementary Fig. 13d)

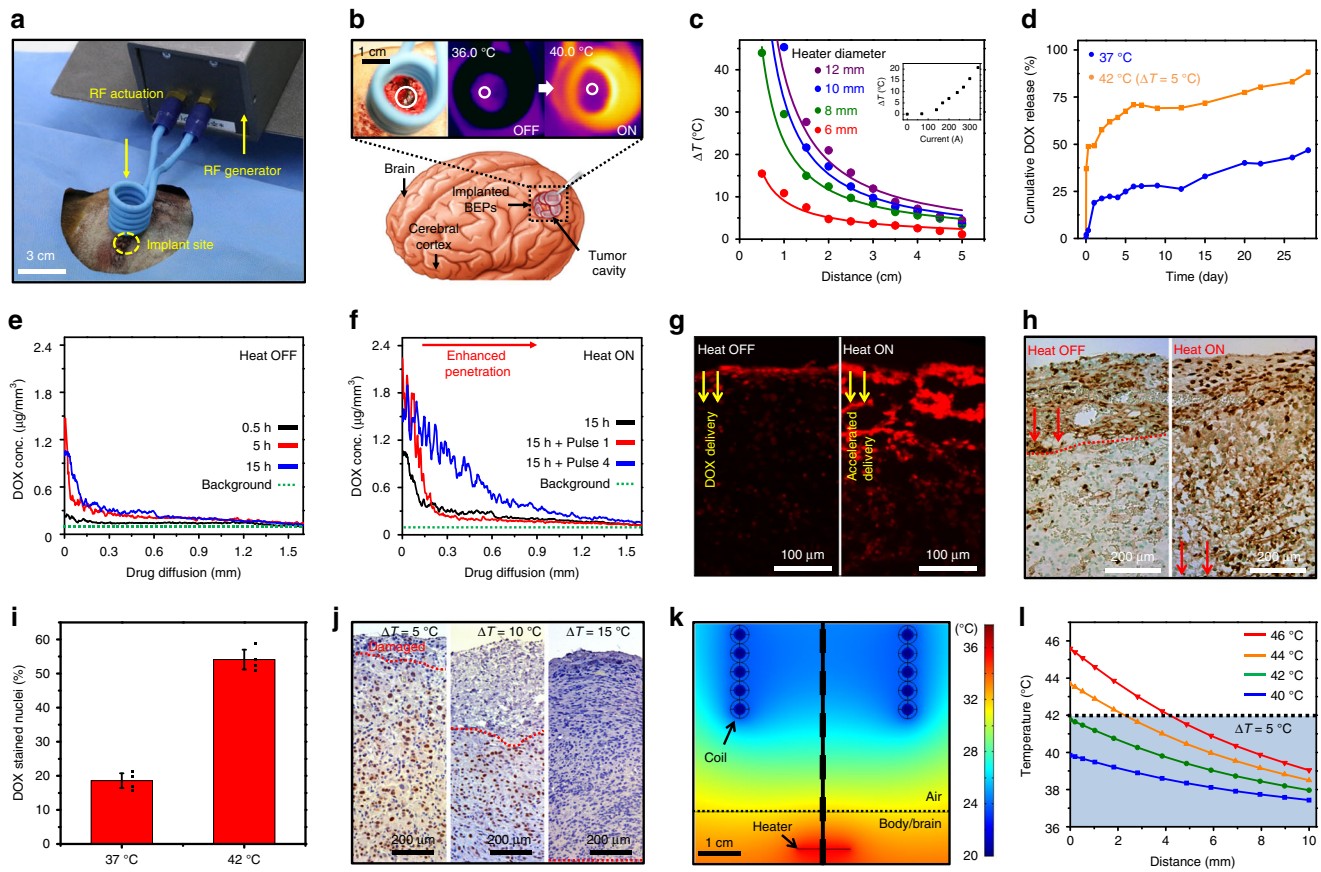

**Fig. 3** Wireless mild-thermic drug delivery. **a** Image of radio frequency (RF) wireless mild-thermic actuation in the canine GBM model. **b** Schematic illustration of implanted BEPs and images of the mild-thermic actuation: optical (left) and infrared camera images before (middle) and during (right) the wireless mild-thermic actuation. **c** Experimentally measured temperature change of the BEP by wireless heating depending on the coil-to-heater distance. The inset shows the temperature change depending on the induced current. **d** Cumulative release of DOX from OST in PBS at 37 °C (blue) and 42 °C (orange) for 4 weeks. **e** In vivo measurement of the DOX concentration at the indicated time points after implantation of the BEP without mild-thermic actuation. **f** In vivo measurement of the DOX concentration at 15 h after implantation of the BEP with the pulsed mild-thermic actuation; control (black; no pulse), 1 pulse (red), or 4 pulses (blue). **g** Fluorescence microscope images after 15 h implantation, which show DOX diffusion from the BEP into U87-MG tumor tissues in the mouse model in vivo without (left) and with (right) the mild-thermic actuation. **h** Terminal deoxynucleotidyl transferase dUTP nick end labeling (TUNEL) assay of U87-MG tumor tissues without (left) and with (right) the mild-thermic actuation. **i** Proportion of DOX-stained cells after exposure to the DOX solution for 1 h at 37 and 42 °C, measured by flow cytometry. Each experiment was repeated at least four times and error bars represent the standard error of the mean value. **j** Effect of elevated temperatures on U87-MG tumor tissues, observed by survivin expression. **k** Contour plot of the temperature distribution during the mild-thermic actuation. **l** Simulated temperature profile from the BEP surface into the brain under various heater temperatures. The shadowed part represents the temperature that can be tolerated by the brain tissue

can be used to optimize the amount of heating. Also, the dimension of the heaters (thickness and diameter) can be decreased or the coil-to-heater distance can be increased further by controlling the instrumental parameters such as the RF frequency (Supplementary Fig. 13e, f). More details are described in Supplementary Methods 1.1.

The amount of the drug release can be increased by mild-thermic actuation at 42 °C ($\Delta T = 5$ °C; Fig. 3d). The concentration gradient of DOX delivered into human GBM tissues (U87-MG) with and without mild-thermic actuation was quantified in an immune-deficient mouse in vivo by integrating equidistant fluorescent signal counts. The natural diffusion of DOX (Fig. 3e) is similar to the natural diffusion of carmustine[41,42], and the mild-thermic actuation dramatically enhances drug diffusion (Fig. 3f, g). The actuation condition of $\Delta T = 5$ °C (30 min of 2 pulses) results in death of deep cancer cells, which is shown in the cross-sectional tissue staining image by terminal deoxynucleotidyl transferase dUTP nick end labeling (TUNEL) assay (Fig. 3h). The drug delivery based on mild-thermic actuation over the longer period of time (42 °C, 2 days, ex vivo) results in the enhanced

drug delivery depth (~11 mm, Supplementary Figs. 14 and 15a), where the drug concentration is over CC50 of the tumor cell (0.15 μg/mL; Supplementary Fig. 15b). This increased drug penetration is due to increased cell membrane permeability[43] at increased temperature. The U87-MG cell is exposed to the drug solution (DOX of 0.2 μg/mL) at 37 °C and 42 °C for 1 h, and the number of nuclei stained by DOX is measured to compare drug diffusion. The three-times higher drug uptake is observed at the higher temperature (Fig. 3i and Supplementary Fig. 16).

The appropriate level of mild-thermic actuation promotes the drug diffusion while minimizing the thermal damage to surrounding brain tissues[44], but excessive hyperthermia may cause apoptosis[45] of normal brain tissues. Thermal damage to the brain tissue is analyzed by detecting expression of apoptosis inhibitors with survivin (Fig. 3j and Supplementary Fig. 17). The temperature increase of 5 °C for 30 min induced minimal apoptosis, while the temperature increase of 10 or 15 °C caused significant cell death (Fig. 3j). Numerical simulations (Fig. 3k, Supplementary Fig. 18 and Supplementary Table 3) also indicate that no excessive heat is applied to the brain tissue

under $\Delta T = 5\,°C$, which is consistent with experiments (Fig. 3j). Heat is dissipated by blood flow at distal regions (Fig. 3k). The absence or presence of skull does not cause differences on temperature distribution according to 3D modeling (Supplementary Fig. 19 and Supplementary Table 4). The details of thermal modeling are described in Supplementary Methods 1.2. Although we tried to maintain the target temperature to be stable as shown in Fig. 3b experiments and Fig. 3k simulations, it is true that temperature fluctuations happen under pulsed wireless thermal actuations in vivo. More reliable and elaborate heating protocols should be developed through large scale in vivo animal experiments in the future.

The optimal coil-to-heater distance at a given heater radius, in which temperature is below $42\,°C$, can be estimated through simulations (Fig. 3l). The drug diffusion coefficient increases significantly by wireless mild-thermic actuation of $\Delta T = 5\,°C$, consistent with previous reports (Supplementary Fig. 18a[43,46]). For experimental validation of the optimal distance, temperature increase by mild-thermic actuation is estimated by either an IR camera or a wireless temperature sensor of the BEP (Supplementary Fig. 20 and Supplementary Methods 1.3). The optimum actuation condition is adopted to the therapy protocol.

**Evaluation of therapeutic efficacy in the mouse and canine GBM model.** We evaluated therapeutic efficacy of the BEP in vivo by using the human xenograft GBM model with immune-deficient mice (BALB/c nude mice). Human GBM cells (U87-MG) were cultured and subcutaneously implanted near the thigh region of 6-week-old nude mice ($n = 33$). Tumor was grown and resected similar to the standard protocol of human brain tumor[47] therapy, and then the BEP of ~14 mm diameter was implanted. Detailed surgical procedures are provided in Supplementary Methods 1.4 and Supplementary Fig. 21. The protocol for the therapy is presented in Fig. 4a. The procedure starts from the tumor resection. Further studies for the nonresectable tumor cases are also needed in the future.

Mice were divided into five groups: each treated with (1) intravenous injection of DOX (IV group, Supplementary Fig. 22a); (2) implantation of the BEP without DOX but with mild-thermic actuation (Heating group; Supplementary Fig. 22b); (3) implantation of the BEP with DOX but without heating (OST group; Supplementary Fig. 22c); (4) implantation of the custom-made control wafer of which the composition is same as the Gliadel wafer (194.4 mg poly[bis(p-carboxyphenoxy)propane] anhydride and sebacic acid containing 7.7 mg carmustine) (control wafer group; Fig. 4b); and (5) implantation of the BEP with DOX and with mild-thermic actuation (OST + Heating group; Fig. 4c). The DOX concentration in the BEP is 0.69 mg/BEP. The Heating and OST + Heating group received wireless mild-thermic treatment for 2 weeks following the therapy protocol (Fig. 4a).

The IV and Heating group exhibited poor prognoses, while the OST and control wafer group showed meaningful suppression of tumor recurrence (Supplementary Table 1). In particular, the OST + Heating group exhibited significantly reduced tumor volume among all groups, even compared with the control wafer group ($p = 0.0048$) as shown in Fig. 4d and Supplementary Table 1. It also leads to the dramatic increase in the survival rate among all groups, even compared with the control wafer group ($p = 0.013$) as shown in Fig. 4e and Supplementary Table 2. This improvement in therapeutic efficacy shows combined benefits of local drug delivery to the target region, long-lasting therapy due to sustained drug release, and enhanced drug penetration by mild-thermic actuation. Since the microenvironment in thigh is different from that in brain, we additionally established a mouse brain tumor model to prove efficacy of mild-thermic actuation in

brain (Supplementary Fig. 23a–e), which exhibited consistent results with the subcutaneous tumor model results (Supplementary Fig. 23 and Supplementary Methods 1.5).

To obtain preclinical data of the BEP in large animals, a canine GBM model was established. Since canine brain is much larger than mouse brain, it allows a similar procedure with standard human GBM surgery (Supplementary Fig. 24). Detailed procedures are described in Supplementary Methods 1.6. Similar to the conventional neurosurgery, the implanted GBM tissues were partially removed by surgery while residual infiltrative tumor tissues remained in the surgical cavity (Fig. 4f). The BEP was implanted to the cavity, and the surgical site was covered by surgical glue and skin without skull. Supplementary Fig. 25 shows that tissues near the cavity have little mechanical damages, confirming the surgical process is successful. The tumor volume continuously increased unless treated (Fig. 4g, left). When wireless mild-thermic treatment ($\Delta T = 5\,°C$, 30 min; 2 pulses) was applied by the BEP, however, tumor growth was suppressed (Fig. 4g, right) by intracranial drug delivery (Fig. 4h).

To evaluate therapeutic effect of the BEP further, both the BEP and a control wafer were implanted to the brain cavity with the remaining brain tumor in two different mongrel dogs. The treatment with the BEP and the control wafer for 2 days caused apoptosis of tumor cells, which are confirmed by TUNEL assay (Fig. 4i, j and Supplementary Fig. 26). Both cases showed apoptosis of tumor cells within 2 mm from the cavity surface. However, the BEP induced apoptosis of deeply invaded microscopic tumor cells (Fig. 4i) at 5 mm (Fig. 4j, Magnified H&E TUNEL images in Supplementary Fig. 26a, b), while the control wafer could not treat tumor cells located over 2 mm depth from the cavity surface (Supplementary Fig. 26c, d). More deeply seated tumor cells could be treated by the BEP, since drug penetration can be extended by mild-thermic actuation of the BEP.

## Discussion
In conclusion, we have developed a flexible, sticky, and biodegradable wireless device using bioresorbable materials and electronics design. The device, together with an associated mild-thermic drug delivery protocol, achieved enhanced therapeutic efficacy in brain tumor treatment. The integrated device and protocol offered wirelessly controlled, spatially focused, and temporally extended delivery of antitumor agents up to deeply located brain tumors. The BEP dramatically suppressed tumor volume and enhanced survival rate in vivo. The fully bioresorbable nature of the BEP provided intracranial biocompatibility and minimized potential side effects. The proposed material and device technology represents an important step toward intracranial treatment of brain tumors.

## Methods
**Fabrication of the BEP.** A sacrificial layer of poly(methyl methacrylate) (PMMA; A11, Microchem, USA) was spin coated (176 g, 30 s) onto a silicon wafer and cured at 180 °C for 3 min. The diluted polyimide (PI) precursor solution (poly(pyromellitic dianhydride-co-4,4′-oxydianiline), amic acid solution; Sigma Aldrich, USA) was spin coated onto the substrate (176 g, 60 s) and cured at 250 °C for 2 h to form a bottom PI layer. PI was diluted by mixing the same mass of the PI precursor solution and 1-methyl-2-pyrrolidinone (Sigma Aldrich, USA). A ZnO thin film (2 nm) was deposited by AC sputtering under an Ar atmosphere (5 mTorr, 30 W) as an adhesion layer. In sequence, 3 μm of magnesium was deposited using a thermal evaporator. Then, AZ5214 photoresist (PR; Microchem, USA) was spin coated and patterned. The magnesium was etched using a custom-made magnesium etchant (nitric acid:deionized water:ethylene glycol; 1:1:3) to pattern the wireless heater and temperature sensor. The diluted PI precursor solution was again spin coated onto the substrate (176 g, 60 s) and cured at 250 °C for 2 h to form a top PI layer. Then, the top and bottom PI layers were patterned using AZ4620 PR (Microchem, USA). The PI was etched by oxygen plasma using a reactive-ion etcher ($O_2$, 100 sccm, 0.1 Torr, 100 W). After patterning the PI, the sacrificial PMMA layer was undercut etched using acetone at 70 °C. The device, delaminated from the substrate, was picked up using a polydimethylsiloxane

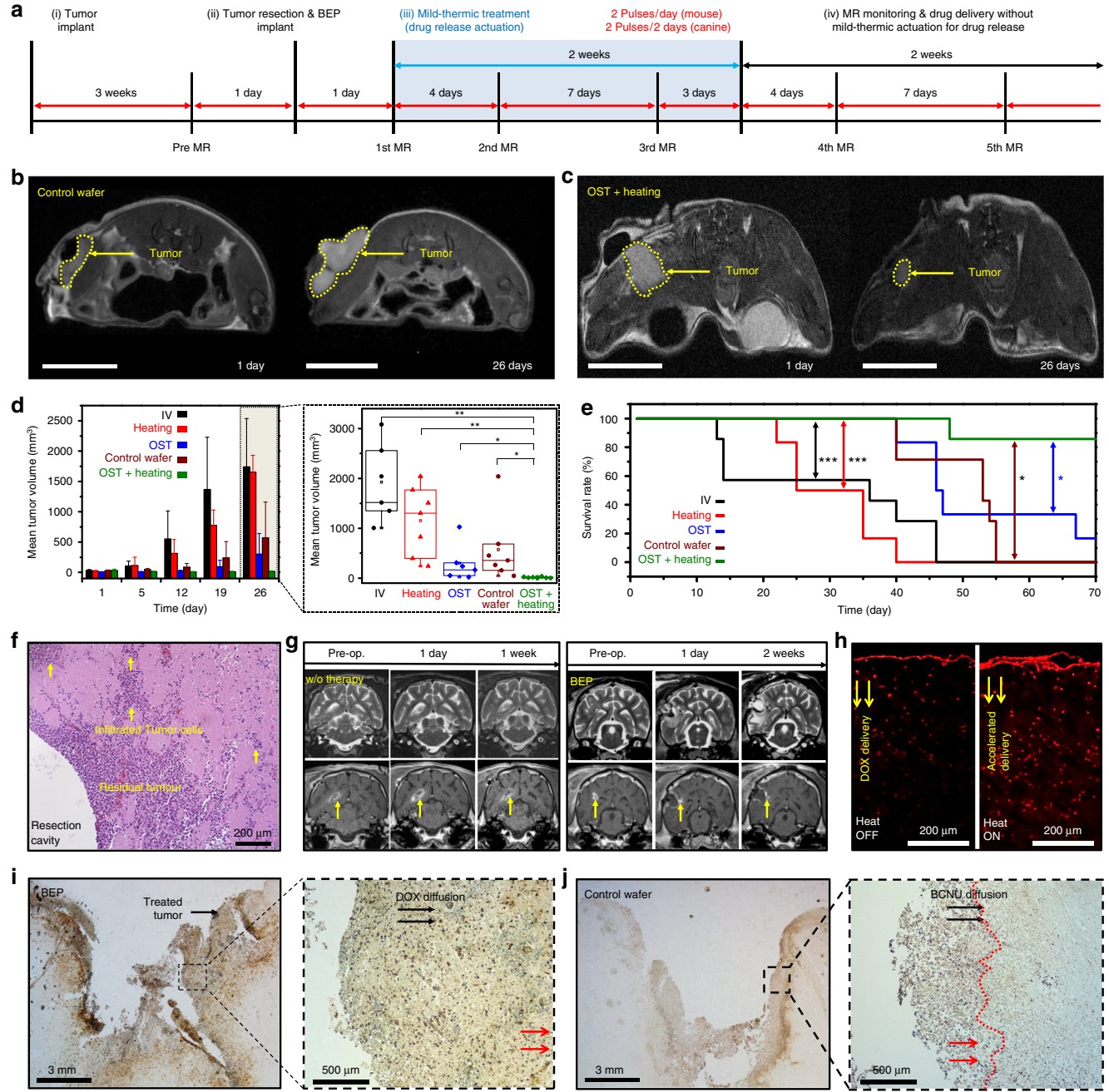

**Fig. 4** Therapeutic efficacy of the BEP in BALB/c nude mouse and canine brain tumor models. **a** Therapy protocol employed to investigate the BEP in both mouse and canine model. **b** Representative axial T2-weighted MR images of the control wafer group, and **c** OST + Heating group at the indicated time points after surgery in the mouse model. **d** Time dependent mean tumor volumes of the indicated groups (left) and box-and-whisker plots of tumor volumes at 26 days after surgery (right). $n = 6, 7, 6, 7, 6$ for IV, Heating, OST, OST + Heating, and control wafer group, respectively in the mouse model. Line: median Box: 25th–75th percentiles, Whisker: min to max, $*p < 0.05$, $**p < 0.01$ by Man–Whitney $U$-test with Bonferroni correction. **e** Kaplan–Meier survival rate plots of the indicated treatment group in the mouse model, $*p < 0.05$ by log-rank test with Bonferroni correction. **f** Histology images of tissues near the J3T-1 implantation site stained with H&E in the canine model. **g** Coronal T2- (top) and contrast-enhanced T1-weighted MR images of the tumor without (left) and with (right) the BEP treatment in the canine model. **h** Fluorescence images of DOX diffused from the BEP into the J3T-1 tissue in the canine model in vivo without (left) and with the mild-thermic actuation in the canine model. **i** TUNEL assay of the BEP treatment case after 2 days from the implantation (left) and its magnified view (right) in the canine model. **j** TUNEL assay of the control wafer treatment case after 2 days from the implantation (left) and its magnified view (right) in the canine model

(PDMS) stamp (Sylgard 184, A:B = 10:1). Then, the bottom PI layer was etched by oxygen plasma. Separately, 3% (w/w) PLA or 8% (w/w) PLGA was spin coated on top of the OST film several times. The patch was exposed to the vapor of boiling chloroform to make the PLA or PLGA sticky prior to transferring the device from the PDMS stamp to the OST film. After the transfer, the top PI layer was etched by oxygen plasma. Finally, PLA and PLGA were spin coated several times for top encapsulation to protect the magnesium-based electronic device. Before

implantation of the BEP, each device that was transferred onto the OST film was cut to have a circular shape with the diameter of 14 mm.

**Mild-thermic actuation procedure**. The mild-thermic actuation was applied under the anesthesia in both mouse and canine model without the skull but with the cover of the surgical glue and skin. The heating procedure consists of 30 min

pulse for two times every day in the mouse model, and 30 min pulse for two times every 2 days in the canine model to prevent anesthetic death.

**Histology**. Tissues were fixed in 10% formalin, incubated in graded ethanol, embedded in paraffin, and cut into 4-μm-thick sections. The histological analyses were performed with H&E staining for basic morphological evaluation and TUNEL (terminal deoxynucleotidyl transferase dUTP nick end labeling) staining for apoptotic cell detection. For immunohistochemistry (IHC), all tissue sections were deparaffinized in xylene and hydrated by immersion in a graded ethanol series. Antigen retrieval was performed in a microwave by placing the sections in epitope retrieval solution (0.01 M citrate buffer, pH 6.0) for 20 min; endogenous peroxidase was inhibited by immersing the sections in 0.3% hydrogen peroxide for 10 min. The sections were then incubated with a primary mouse monoclonal antibody to macrophage (ThermoFisher Scientific, USA, MAC387, dilution 1:200) for inflammatory macrophages in a canine model or a rabbit monoclonal antibody to survivin (Cell Signaling, USA, #2808, dilution 1:300) in Dako REAL antibody diluent (Dako, USA) for a mouse model. The IHC staining of GFAP and Iba-1 for biocompatibility test was performed with the rabbit monoclonal anti-GFAP (Abcam, USA, ab7260, dilution 1:2000) and anti-Iba-1 (Wako Chemicals, USA, 019–19741, dilution 1:3000) using the Dako Autostainer Link 48 system according to recommended protocols. Two regions of interest from each slide were acquired and all images were analyzed using Image J.

**Statistics**. The final tumor volume for all mice in each group was compared and evaluated by using the Mann–Whitney $U$-test. The $p$ value for the individual test was multiplied by the number of comparison made (Bonferroni correction). The log-rank test was used to compare the survival plot of each group. The $p$ value for the individual test was also multiplied by the number of comparison made (Bonferroni correction). The paired $t$-test was used to compare the rotarod retention time of two groups. The paired $t$-test was used to compare the quantification results of immunohistochemistry.

**Ethical approval**. This study was approved by our Institutional Animal Care and Use Committee (IACUC; No. 14-0156-C2A3) and was performed in accordance with our IACUC guidelines and with the National Institute of Health Guide for the Care and Use of Laboratory Animals.

**Cell lines**. The human GBM cell line (U87-MG) was obtained from the American Type Culture Collection (Rockville, USA, HTB-14) and maintained in RPMI medium with 10% fetal bovine serum (FBS) at 37 °C. All cell lines were routinely tested to exclude infection with mycoplasma and reauthenticated using microsatellite profiling immediately prior to manuscript submission. The canine glioblastoma cell line (J3T-1), which was derived from the parental canine glioma cell line (J3T), was provided by T.I, Okayama University Graduate School of Medicine, Dentistry, and Pharmaceutical Sciences (Japan) after obtaining research ethics approval, and maintained in the Roswell Park Memorial Institute (RPMI) medium with 10% FBS at 37 °C[48–50]. For this study, the cell line used was routinely tested to exclude infection with mycoplasma.

**Reporting summary**. Further information on research design is available in the Nature Research Reporting Summary linked to this article.

## Data availability
The datasets generated during and/or analysed during the current study are available from the corresponding author upon reasonable request.

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

## Acknowledgements

This work was supported by IBS-R006-A1 and IBS-R006-D1. This work was also supported by Creative-Pioneering Researchers Program through Seoul National University (SNU). N.L., L.W. and S.Q. acknowledge the financial support from the US Office of Naval Research (ONR) under Grant No. N00014-16-1-2044. The Korea Basic Science Institute (Seoul) is acknowledged for the HPLC data.

## Author contributions

J.L., H.R.C., G.D.C. and H.S. designed the experiments. J.L., H.R.C., G.D.C., H.S., S.L., C.-K.P., J.W.K., S.Q., L.W., D.K., T.K., J.K., H.L., W.L. and S.H.C. performed experiments and analysis. S.L. and S.H.C. established the animal models. S.K. performed the electromagnetic simulation and analysis. S.-T.L. analyzed the electrophysiology data. T.I. provided the canine glioblastoma cell line (J3T-1). J.L., H.R.C., G.D.C., H.S., N.L., T.H., S.H.C. and D.-H.K wrote the paper.

## Competing interests

The authors declare no competing interests.

## Additional information

**Peer Review Information** *Nature Communications* thanks Henry Brem, Martine Roussel and the other, anonymous, reviewer for their contribution to the peer review of this work. Peer reviewer reports are available.

