## [Peer Review File · Nature Communications]

Reviewers' comments:

Reviewer #1 (Remarks to the Author):

The authors present work on developing a biodegradable wireless device for drug delivery intended for the treatment of brain tumors. The authors show through extensive in vitro and in vivo models that their device with thermic actuation can deliver doxorubicin at sufficient quantities to effect tumor growth in a flank murine model and canine brain tumor model. They also present behavioral and safety data. Additionally the MRI compatibility of the BEP is a positive aspect of the device. We approve this manuscript's acceptance for publication once some concerns, outlined below, are addressed.

We would suggest that the authors update their references for Gliadel and include the 2 meta-analyses published within the last 4 years: Xing WK et al. Drug Des Devel Ther. 2015, PMID: 26170620 and Chowdhary SA et al. J Neurooncol. 2015, PMID: 25630625

The authors include a lot of cortical mapping references and might want to include references to other efficacy studies with doxorubicin and the same U87 tumor model. It is also striking that the authors do not cite previously related work from PNAS which shows doxorubicin and temozolomide released from a microcapsule in the brain (Upadhyay, 2014). Additionally some references are old and should be updated.

In reference to the unidirectional release of drug away from the CSF (Lines 135). It is not correct to state that the unidirectional manner of the BEP can localize the drug diffusion only to certain brain tissue and not CSF when the device is implanted intracranially.

The text needs minor editing for English, and there are also some typos within, i.e., Line 113- 'sustrained- sustained'

For the sheer stress test bovine muscle is unlike brain tissue so the adhesion of the OST film may very well be different in the brain. This should be stated.

While Figure 2c shows some diffusion the limitations should be stated.

What is the maximum loading capacity of DOX in the BEP?

For the thermal effect on DOX release (Fig 3d), that was constant 37 or 42C. Presumably the RF would not be constant for an implant, so there might be bursts of release of the DOX, correct? Pulse 1 and pulse 4? While thermal actuation is simulated in this paper it should be stated that it would have to be carried out to be shown and proven.

The statement on lines 143-145 should include a statement that there remains the need to do further studies.

Figure 3i should be detailed further. What do the 4 colored lines represent?

Figure 4 should state which animal model is being referred to in all images (4a-4j). It is confusing. Also does the time table refer to just the mouse model or both models? Please clarify.

What do the authors mean in line 244- the DOX concentration is set to exhibit similar therapeutic effect to the control wafer? Does the control wafer include carmustine? The DOX-BEP should have the maximally tolerated dose of DOX, not carmustine.

For Figure 4E does the timeline start at tumor implantation or tumor resection? If the Control wafer contains carmustine, where is the untreated control for the experiment? Were the people who resected the tumor and placed the animals into groups and the people who measured tumor volume blinded to the groups?

The authors should cite a Gliadel reference or preclinical Gliadel reference instead of Ref 17.

For the canine study how many pulses total were given? How many dogs total? Was the "control polymer" a polymer containing carmustine?

An experiment with unresected murine tissue would be interesting to see if the DOX-BEP could have efficacy against a larger tumor burden. The experiment could be started 5-10 days after initial tumor injection. Also there would be less variability without tumor resection. Were the surgeons and subsequent device implanters blinded to the group/device type they were implanting? Was it the maximum DOX payload that can be loaded into the device? Is 0.69mg DOX fully loaded - 100% loading capacity? What is the maximum payload?

How far into the brain (in mm) can the BEP be placed and still have communication with the RF?

Supplementary Information 1.1- two of the equations are hard to read

For section Suppl section 2.7- did this method of fluorescence microscopy apply to all in vivo specimens?

Please include the depth of injection for the intracranial murine tumor placement.

Suppl Figure 16b seems to be mislabeled. 16c appears to be the magnified version not 16b.

Fig 18c is 5mm of human skull? Please clarify.

In Suppl Fig 22d it appears that the wafer is sitting on top of the skull- Is the wafer placed intracranially or above the skull? Also please state how the mean tumor volumes calculated in (j).

Reviewer #2 (Remarks to the Author):

In this manuscript, Lee and collaborators describe a new flexible, biodegradable, wireless, and activated by mild heat device for drug delivery to treat brain tumors. The authors show that the drug can be delivered to the tumor site without leakage. The device completely dissolved without debris left behind by being converted to biocompatible metabolites, within 10 weeks in the canine brain, a time that corresponds to the entire time of drug release. No inflammatory responses were detected. Finally, the authors provide data assessing the efficacy of the device in vivo. Experiments and results are well and clearly described. If efficacious in treating patients with brain tumors, it certainly be a great improvement compared to current therapeutic approaches. I

Criticisms:

Because treatment with a single drug has proven to be ineffective, would the device be used to deliver combination therapies? Does the vehicle used for drug resuspension matters to keep the integrity of the device? The authors might want to address these two questions in the Discussion.

Results in Figure 2 h and I are difficult to see. Quantification of the immunohistochemistry would help.

The authors use the word actuation throughout the manuscript. Do the authors mean activation as in “mild-thermic activation” instead of “mild-thermic actuation”?

Reviewer #3 (Remarks to the Author):

In their manuscript “Flexible, Sticky, and Biodegradable Wireless Device for Drug Delivery to Brain Tumours” Lee, Cho, Cha and colleagues discuss the development of a bioresorbable drug-delivery patch thermally driven by eddy currents triggered within an integrated metal patch by an externally applied radiofrequency alternating magnetic field. The team evaluated their device in xenograft models of glioblastoma (GBM) in immunodeficient mice and dogs and corroborated the enhanced ability to deliver doxorubicin (DOX) to the tumor resection sites. The latter resulted in improved survival rates in the mouse GBM model. While ambitious, innovative, and extensive in its scope this study leaves a few technical issues unaddressed and could be further improved through additional analyses. My specific comments are detailed below:

1. The use of eddy current heating to facilitate the DOX delivery to the tumor resection site is an interesting concept. The technical description of the magnetic field conditions and the overall device design, however, is somewhat nebulous. The authors use position of the driving field coil with respect to the eddy heater as a key parameter of their design. This is meaningless unless precisely the same coil is applied for all future applications of this technology. The latter is unlikely given the fact that the field coil and the driving electronics are designed for induction welding and are not optimized for biomedical purposes. The authors should quantify the amplitude of the magnetic field rather than distance and calculate the resulting eddy currents in their heater.

2. They should also elaborate on the principles underlying the design of the eddy heater – what were the key optimization criteria? Would a field with a higher amplitude or frequency afford further miniaturization of the heater? The calibration curve for the temperature as a result of eddy current should be provided (T vs. I or T vs. H).

3. The authors emphasize the degradable nature of their patch. Simultaneously they design the patch to be unidirectionally functional, releasing drugs toward the brain while avoiding excessive leakage into the epidural or subdural space. I am very perplexed by this concept. First, the tumor cavity is unlikely to have a uniform shape enabling precise placement of the patch in a manner that only permits release into the brain. Second, as the device degrades, the drug is expected to leak essentially everywhere in the cavity. The authors should show profiles of the drug leakage overlaid with device degradation – immunohistochemistry with the device still in place over an entire operation window would be most enlightening.

4. The authors perform their adhesion tests on the bovine muscle tissue. But the intended application is the surface of the brain. The two types of tissue have rather different mechanical properties and it is unclear why the bovine muscle was used as a testbed. Adhesion tests on the actual mouse or canine brain would be more representative.

5. Immunohistochemistry analyses of the foreign body response are somewhat qualitative. The data should be shown for multiple animals (typically n=5-8) for several time points within the lifespan of the device. These data should be collected in the brain of the same mouse models used for GBM evaluation experiments.

6. The authors discuss using EEGs as a means to confirm the “normal” state of the brain tissue following implantation of their patches. This entire part of the manuscript is rather confusing. First, looking at the electrode positioning within the brain, it appears that the authors used penetrating electrodes placed within the brain tissue following craniotomy. A typical EEG is performed using electrodes placed on the surface of the skull. Second, it is unclear how the data was recorded and filtered. Supplementary Figure 9c does not appear to have any scale bars on either axis. It not even clear what's on the axes. Third, what features did the authors use to confirm that the electrophysiological activity is “normal”. What constitutes normal for this particular electrode placement? What would be a signature of an abnormality?

7. It is unclear what motivated the choice of the rotarod motor performance as a metric to corroborate the device biocompatibility. Was the device implanted above the motor cortex?

Reviewer #1:

Summary Comments: *The authors present work on developing a biodegradable wireless device for drug delivery intended for the treatment of brain tumors. The authors show through extensive in vitro and in vivo models that their device with thermic actuation can deliver doxorubicin at sufficient quantities to effect tumor growth in a flank murine model and canine brain tumor model. They also present behavioral and safety data. Additionally the MRI compatibility of the BEP is a positive aspect of the device. We approve this manuscript's acceptance for publication once some concerns, outlined below, are addressed.*

Our response: We are grateful for the reviewer's precise summary and kind recommendation to publish in *Nature Communications*. We have revised the manuscript in a point-by-point manner below according to the reviewer's insightful comments.

Comment #1: *We would suggest that the authors update their references for Gliadel and include the 2 meta-analyses published within the last 4 years: Xing WK et al. Drug Des Devel Ther. 2015, PMID: 26170620 and Chowdhary SA et al. J Neurooncol. 2015, PMID: 25630625. The authors include a lot of cortical mapping references and might want to include references to other efficacy studies with doxorubicin and the same U87 tumor model. It is also striking that the authors do not cite previously related work from PNAS which shows doxorubicin and temozolomide released from a microcapsule in the brain (Upadhyay, 2014). Additionally some references are old and should be updated.*

Our response: We appreciate the reviewer for the comments on the references. We removed old references and added new references including recommended ones to introduce the recent progress in this field.

Our modification: (Line 25, page 2: in revised main text)

"...since the blood-brain barrier (BBB) blocks delivery of drugs to brain tumours⁴. Several researches have improved controlled and targeted drug delivery to brain tumours⁵⁻⁷. For example, the biodegradable polymeric wafer^{8,9} (Gliadel wafer, Arbor Pharmaceuticals, USA) implanted near the brain surgery site locally delivers drugs to remaining brain tumours and exhibits meaningful improvement..."

(Line 2, page 7: in revised main text)

"...This shows minimal DOX leakage to CSF, which is important to prevent drug waste and potential side-effects³⁷..."

(Reference: in revised main text)

~~7. Westphal, M. et al. A phase 3 trial of local chemotherapy with biodegradable carmustine (BCNU) wafers (Gliadel wafers) in patients with primary malignant glioma. *Neuro-Oncology* **5**, 79-88 (2003)-33. Bleyer, W.A., Drake, J.C. & Chabner, B.A. Neurotoxicity and Elevated Cerebrospinal Fluid Methotrexate Concentration in Meningeal Leukemia. *New Eng. J. Med.* **289**, 770-773 (1973).~~

5. Upadhyay, U.M. et al. Intracranial microcapsule chemotherapy delivery for the localized treatment of rodent metastatic breast adenocarcinoma in the brain. *Proc. Nat. Acad. Sci.* **111**, 16071-16076 (2014).

8. Xing, W.-k., Shao, C., Qi, Z.-y., Yang, C. & Wang, Z. in *Drug. Des. Devel. Ther.* **9**, 3341-3348 (2015).

9. Chowdhary, S.A., Ryken, T. & Newton, H.B. Survival outcomes and safety of carmustine wafers in the treatment of high-grade gliomas: a meta-analysis. *J. Neurooncol.* **122**, 367-382 (2015).

37. Tangpong, J. et al. Doxorubicin-induced central nervous system toxicity and protection by

xanthone derivative of *Garcinia mangostana*. *Neuroscience* **175**, 292-299 (2011).

Comment #2: *In reference to the unidirectional release of drug away from the CSF (Lines 135). It is not correct to state that the unidirectional manner of the BEP can localize the drug diffusion only to certain brain tissue and not CSF when the device is implanted intracranially.*

Our response: We thank for the valuable comment. As the reviewer pointed out, although the drug leakage can be decreased, the perfect localization of the drug delivery cannot be achieved. We modified texts to clarify this point.

Our modification: (Line 18, page 6: in revised main text)

“...suppresses the drug release to the PBS solution dramatically (Fig. 2d red). **The selective coating of PLA only on the top surface of the device where the device is exposed to CSF can decrease the unintended drug leakage to other regions (Supplementary Fig. 5 and 6; *in vitro* and *in vivo*, respectively), although perfect prevention of the drug diffusion to the CSF cannot be achieved...**”

(Line 19, page 4: in revised main text)

“...These conformal and strong adhesion improves the **efficiency** of **the** drug delivery. Meanwhile, the hydrophobic PLA top encapsulation **reduces** undesirable drug delivery to CSF...”

Comment #3: *The text needs minor editing for English, and there are also some typos within, i.e., Line 113- ‘sustrained- sustained’*

Our response: We appreciate the reviewer for pointing out typos and the editing issue. We corrected the typos and edited other minor issues. The modifications are marked in red.

Our modification: (Line 10, page 5: in revised main text)

“... (e.g., doxorubicin; DOX) (Fig. 2a, green), and thus enables good adhesion and **sustained** drug release...”

sustrained → sustained

(Line 4, page 2: in revised main text)

“...The requirements of the device for the **controlled** intracranial drug delivery to brain tumours are as...”

controleed → controlled

(Line 12, page 3: in revised main text)

“...Toward accomplishment of these challenging goals **altogether**, a novel soft biodegradable...”

all together → altogether

(Line 19, page 3: in revised main text)

“...delivery duration. **The** flexibility of the oxidized-starch-based (OST-based) patch and its **hydrophilic**/hydrophobic bifacial design allow conformal adhesion to the target brain ...”

hydrophillic → hydrophilic

(Line 21, page 3: in revised main text)

“...and sustained drug delivery, while **reducing** unintended drug release to CSF. Fully bioresorbable and...”

reduce → reducing

(Line 20, page 4: in revised main text)

“...drug delivery to CSF. The **imine** conjugation also helps long-term sustained drug delivery...”
immine → imine

(Line 21, page 7: in revised main text)

“...affect mouse brain functions. We evaluated **the** whether the BEP can affect the behavior...”

(Line 7, page 8: in revised main text)

“...Supplementary Fig. 12a). **Also**, the intracranial implantation of the BEP to canine brain did not...”
Also → Also,

(Line 5, page 13: in revised main text)

“...The integrated device and protocol **offered** wirelessly-controlled, spatially-focused, and...”
offerrred → offered

(Line 25, page 10: in revised Supplementary Information)

“...This regimen was previously used in an intraparenchymal and **cavernous** sinus tumour model...”
carvernous → cavernous

(Line 5, page 17: in revised Supplementary Information)

“...two consecutive revolutions. At each test **day**, mice were tested for three trials, and the mean latencies...”
days → day

(Supplementary Fig. 23b: in revised Supplementary Information)

weeks → week

Comment #4: *For the shear stress test bovine muscle is unlike brain tissue so the adhesion of the OST film may very well be different in the brain. This should be stated.*

Our response: We thank the reviewer for raising this issue. The mechanical strength of the brain tissue [1] is much lower than the adhesion force between the BEP and brain. Therefore, the brain tissue becomes mechanically torn before the detachment of the device from the tissue surface in the adhesion test [2]. Although the bovine muscle is different from the brain tissue, its mechanical strength is stronger than the brain tissue, and thus the adhesion test could be proceeded, which showed the strong adhesion of the patch to the brain tissue indirectly. We added this point in the revised manuscript.

[1] Budday, S. *et al.* Mechanical properties of gray and white matter brain tissue by indentation. *J. Mech. Behav. Biomed.* **46**, 318-330, (2015).

[2] Yuk, H., Zhang, T., Lin, S., Parada, G. A. & Zhao, X. Tough bonding of hydrogels to diverse non-

porous surfaces. *Nat. Mater.* **15**, 190, (2015).

Our modification: (Line 21, page 5: in revised main text)

“...Although we loaded two kinds of drugs, the integrity of the device was well maintained. Since the adhesion force between the OST film and the brain tissue is stronger than the mechanical strength of the brain tissue^{34, 35}, the brain tissue is mechanically torn before the detachment of the film from the brain surface. Therefore, the effect of oxidization of starch on the adhesion strength was indirectly tested on the bovine muscle instead of the brain tissue, since the bovine muscle has higher mechanical strength than the brain tissue. The shear adhesion test (Fig. 2b inset) of OST exhibited its strong adhesion to the muscle tissues (Fig. 1a right) than that of the unmodified starch film...”

(Reference: in revised main text)

34. Budday, S. *et al.* Mechanical properties of gray and white matter brain tissue by indentation. *J. Mech. Behav. Biomed. Mater.* **46**, 318-330 (2015).
35. Yuk, H., Zhang, T., Lin, S., Parada, G.A. & Zhao, X. Tough bonding of hydrogels to diverse non-porous surfaces. *Nat. Mater.* **15**, 190 (2015).

Comment #5: While Figure 2c shows some diffusion the limitations should be stated

Our response: We thank the reviewer for raising this issue. Although some diffusion is shown in Fig. 2c, the diffusion length may not be enough, which can be increased by additional mild-thermic actuations. We modified the manuscript accordingly.

Our modification: (Line 5, page 6: in revised main text)

“...to the convoluted brain surface by changing the ratio of OST to glycerol (Supplementary Fig. 4). The strong adhesion and softness enable the conformal contact of the BEP at the tissue surface (Fig. 2c), which allows local drug delivery (Fig. 2c inset, white circle). The diffusion length by the natural diffusion (Fig. 2c) is small. However, it can be increased by additional mild-thermic actuations...”

Comment #6: What is the maximum loading capacity of DOX in the BEP?

Our response: We appreciate the reviewer for the question. DOX is highly miscible to the OST. The maximum DOX loading amount which we have observed was 6.831 mg per 1 BEP with a diameter of 1.4 mm. The total amount of DOX delivered to the brain can be increased further by using multiple BEPs. We modified the text accordingly.

Our modification: (Line 14, page 5: in revised main text)

“...outstanding therapeutic effect and easy visualization using fluorescence. The maximum DOX loading that we have observed was 6.831 mg per one BEP since the DOX is highly miscible to the OST. The total amount of DOX can be increased further by using multiple BEPs...”

Comment #7: For the thermal effect on DOX release (Fig 3d), that was constant 37 or 42C. Presumably the RF would not be constant for an implant, so there might be bursts of release of the DOX, correct? Pulse 1 and pulse 4? While thermal actuation is simulated in this paper it should be stated that it would have to be carried out to be shown and proven.

Our response: We thank the reviewer for the comment. The temperature for the DOX release test *in vitro*

in Fig. 3d was maintained to be constant, 37 and 42 °C. For experiments *in vivo*, 1 pulse (30 min) or 4 pulses (2 h) were applied. Although we tried to maintain the target temperature to be stable *in vivo* as shown in Fig. 3b experiments and Fig. 3k simulations, it is true that temperature fluctuations should happen under the *in vivo* conditions. We added this point in the revised manuscript. More reliable heating protocols should be developed through large scale *in vivo* animal experiments in the future.

Our modification: (Line 9, page 10: in revised main text)

“...The details of thermal modeling are described in Supplementary Text 1.1. Although we tried to maintain the target temperature to be stable as shown in Fig. 3b experiments and Fig. 3k simulations, it is true that temperature fluctuations happen under pulsed wireless thermal actuations *in vivo*. More reliable and elaborate heating protocols should be developed through large scale *in vivo* animal experiments in the future...”

Comment #8: The statement on lines 143-145 should include a statement that there remains the need to do further studies.

Our response: We are grateful for the reviewer’s comment. We studied the tissue response up to 10-weeks post-surgery. As the reviewer pointed out, however, further studies for the longer-term observation are needed. We modified the text accordingly.

Our modification: (Line 12, page 8: in revised main text)

“...physiological complications were not observed (Supplementary Fig. 12b). The longer-term studies to observe the effect of hydrolyzed materials *in vivo* are needed in the future...”

Comment #9: Figure 3l should be detailed further. What do the 4 colored lines represent?

Our response: We thank the reviewer for pointing out this issue. Figure 3l represents the temperature distribution around the wireless heater in the brain under various heater temperatures. Each color represents the different heater temperature. We modified the figure and caption for clarification.

Our modification: (Figure 3l: in revised main text)

“I Simulated temperature profile from the BEP surface into the brain under various heater temperatures.”

Comment #10: Figure 4 should state which animal model is being referred to in all images (4a-4j). It is confusing. Also does the time table refer to just the mouse model or both models? Please clarify.

Our response: We thank the reviewer for the comment. Figure 4b-e are for the mouse model, and Fig.

4f-j are for the canine model. We modified Fig. 4a to address this information in the timetable clearly. We also clarified this point in the figure caption accordingly.

Our modification: (Fig. 4a: in revised main text)

(Fig. 4: in revised main text)

Fig. 4 Therapeutic efficacy of the BEP on subcutaneous brain tumour models in BALB/c nude mouse and canine brain tumour models. **a** Therapy protocol employed to investigate the BEP in both mouse and canine model. **b** Representative axial T2-weighted MR images of the ‘Control wafer’ group, and **c** ‘OST+Heating’ group at the indicated time-points after surgery in the mouse model. **d** Time dependent mean tumour volumes of the indicated groups (left) and box-and-whisker plots of tumour volumes at 26 days after surgery (right). n = 6, 7, 6, 7, 6 for IV, Heating, OST, OST+Heating, and Control wafer group, respectively in the mouse model. **e** Kaplan-Meier survival rate plots of the indicated treatment group in the mouse model. **f** Histology images of tissues near the J3T-1 implantation site stained with H&E in the canine model. **g** Coronal T2- (top) and contrast-enhanced T1-weighted MR images of the tumour without (left) and with (right) the BEP treatment in the canine model. **h** Fluorescence images of DOX diffused from the BEP into the J3T-1 tissue in the canine model *in vivo* without (left) and with the mild-thermic actuation in the canine model. **i** TUNEL assay of the BEP treatment case after 2 days from the implantation (left) and its magnified view (right) in the canine model. **j** TUNEL assay of the control wafer treatment case after 2 days from the implantation (left) and its magnified view (right) in the canine model.

Comment #11: What do the authors mean in line 244- the DOX concentration is set to exhibit similar therapeutic effect to the control wafer? Does the control wafer include carmustine? The DOX-BEP should have the maximally tolerated dose of DOX, not carmustine.

Our response: The control wafer contains carmustine, whose composition is same as the commercially available Gliadel wafer®. In case of the BEP, it contains DOX instead of Carmustine whose amount is 0.69 mg per BEP. We revised the text to clarify this information.

Our modification: (Line 10, page 11: in revised main text)

“...(4) implantation of the custom-made control wafer of which the composition is same as the Gliadel wafer (194.4 mg poly[bis(p-carboxyphenoxy)propane] anhydride and sebacic acid containing 7.7 mg carmustine) (Control wafer group; Fig. 4b); and (5) implantation of the BEP with DOX and with mild-thermic actuation (OST+Heating group; Fig. 4c). The DOX concentration in the BEP is 0.69 mg/BEP. The ‘Heating’ and ‘OST+Heating’ group received wireless...”

Comment #12: For Figure 4E does the timeline start at tumor implantation or tumor resection? If the Control wafer contains carmustine, where is the untreated control for the experiment? Were the people who resected the tumor and placed the animals into groups and the people who measured tumor volume blinded to the groups?

Our response: We thank the reviewer for the comment. The timeline starts from the tumour resection. Also, all people who resected the tumour, placed the animal, and measured the tumour volume were blinded each other. We compared the BEP with the ‘IV’ group. Even with IV injection of DOX, which is an additional treatment than the negative control, ‘IV’ group exhibited significantly increased tumour volume compared to the positive control groups (OST, OST+Heating). We clarified these points in the revised manuscript.

Our modification: (Line 4, page 11: in revised main text)

“...The protocol for the therapy is presented in Fig. 4a. **The procedure starts from the tumour resection...**”

(Line 13, page 7: in revised Supplementary Information)

“...and was performed in accordance with our IACUC guidelines and with the National Institute of Health Guide for the Care and Use of Laboratory Animals. **The researchers who carried out surgery, managed the animal, and measured the tumour volume were blinded each other...**”

Comment #13: *The authors should cite a Gliadel reference or preclinical Gliadel reference instead of Ref 17.*

Our response: We thank the reviewer for the comment. We replaced reference 17 to the preclinical Gliadel reference.

Our modification: (Reference: in revised main text)

~~17. Rosenblum, D., Joshi, N., Tao, W., Karp, J.M. & Peer, D. Progress and challenges towards targeted delivery of cancer therapeutics. *Nat. Commun.* **9**, 1410 (2018).~~

20. Zhao, M. *et al.* Post-resection treatment of glioblastoma with an injectable nanomedicine-loaded photopolymerizable hydrogel induces long-term survival. *Int. J. Pharm.* **548**, 522-529 (2018).

Comment #14: *For the canine study how many pulses total were given? How many dogs total? Was the “control polymer” a polymer containing carmustine?*

Our response: 2 pulses per every 2 days were applied for the canine study, and 2 dogs were used for each group. ‘Control polymer’ is composed of 194.4 mg poly[bis(p-carboxyphenoxy)propane] and 7.7 mg carmustine. We clarified these points in the revised manuscript.

Our modification: (Line 10, page 11: in revised main text)

“...implantation of the **custom-made control wafer of which the composition is same as the Gliadel wafer (194.4 mg poly[bis(p-carboxyphenoxy)propane] anhydride and sebacic acid containing 7.7 mg carmustine) (Control wafer group; Fig. 4b)...**”

(Line 17, page 10: in revised Supplementary Information)

“...**Canine brain tumour implantation. 2 Adult male mongrel dogs per group** (International Laboratory Animal Center, Korea) weighing approximately 10–12 kg were used in this study. ...”

(Fig. 4a: in revised main text)

Comment #15-1: An experiment with unresected murine tissue would be interesting to see if the DOX-BEP could have efficacy against a larger tumor burden. The experiment could be started 5-10 days after initial tumor injection. Also there would be less variability without tumor resection.

Our response: We thank the reviewer for the suggestion of the animal experiment with unresected tumour. Since the standard treatment protocol includes the tumour resection which has dramatically improved the therapeutic effect [3], our experiment was designed to include the tumour-resection procedure to set the condition as similar as the standard treatment procedure. However, the suggestion for the study regarding non-resectable brain tumours would be meaningful. We will be able to study the non-resectable tumour cases in the future studies. We added this point in the manuscript.

[3] Brown, T.J. *et al.* Association of the extent of resection with survival in glioblastoma: A systematic review and meta-analysis. *JAMA Oncol.* **2**, 1460-1469 (2016).

Our modification: (Line 1, page 11: in revised main text)

“...subcutaneously implanted near the thigh region of 6-week-old nude mice (n = 33). Tumour was grown and resected similar to the standard protocol of human brain tumour⁴⁸ therapy, and then the BEP of ~14 mm diameter was implanted. Detailed surgical procedures are provided in Supplementary Text 1.3 and Supplementary Fig. 21. The protocol for the therapy is presented in Fig. 4a. The procedure starts from the tumour resection. Further studies for the non-resectable tumour cases are also needed in the future...”

(References: in revised main text)

47. Brown, T.J. *et al.* Association of the extent of resection with survival in glioblastoma: A systematic review and meta-analysis. *JAMA Oncol.* **2**, 1460-1469 (2016).

Comment #15-2: Were the surgeons and subsequent device implanters blinded to the group/device type they were implanting?

Our response: All the people who resected the tumour, implanted the device, placed the animal, and measured tumour volume were blinded each other. We modified the text to clarify this point.

Our modification: (Line 13, page 7: in revised Supplementary Information)

“...and was performed in accordance with our IACUC guidelines and with the National Institute of Health Guide for the Care and Use of Laboratory Animals. The researchers who carried out surgery, managed the animal, and measured the tumour volume were blinded each other...”

Comment #15-3: Was it the maximum DOX payload that can be loaded into the device? Is 0.69mg DOX fully loaded - 100% loading capacity? What is the maximum payload?

Our response: We appreciate the reviewer for the question. DOX is highly miscible to the OST. The

maximum DOX loading amount which we have observed was 6.831 mg per 1 BEP with a diameter of 1.4 mm. The total amount of DOX delivered to the brain can be increased further by using multiple BEPs. We modified the text accordingly.

Our modification: (Line 14, page 5: in revised main text)

“...outstanding therapeutic effect and easy visualization using fluorescence. **The maximum DOX loading that we have observed was 6.831 mg per one BEP since the DOX is highly miscible to the OST. The total amount of DOX can be increased further by using multiple BEPs. Further optimization to increase the drug loading amount should be done in the future.** Temozolomide...”

Comment #16: How far into the brain (in mm) can the BEP be placed and still have communication with the RF?

Our response: We thank the reviewer for the comment. The maximum distance within which the wireless mild-thermic actuation can be applied is ~30 mm (Figure 3c). By changing the frequency and coil design, further distances can also be covered (Fig. 3c and Supplementary Figure 13).

Our modification: (Line 4, page 9: in revised main text)

“...can be used to optimize the amount of heating. **Also, the dimension of the heaters (thickness and diameter) can be decreased or the coil-to-heater distance can be increased further by controlling the instrumental parameters such as the RF frequency (Supplementary 13e and f). More details are described in Supplementary Text 1.2....**”

Comment #17: Supplementary Information 1.1- two of the equations are hard to read.

Our response: We thank the reviewer for the comment, and we modified the equations for better readability.

Our modification: (Line 25, page 3: in revised Supplementary Information)

$$\rho_c = \rho_0 [1 + \alpha_T (T - T_0)]$$

$$Q_0 = \frac{M_l C_p (T_{in} - T)}{2\pi r A}$$

Comment #18: For section Suppl section 2.7- did this method of fluorescence microscopy apply to all in vivo specimens?

Our response: We used the same method to all *in vivo* specimens.

Our modification: (Line 5, page 15: in revised Supplementary Information)

“...camera (DFC365 FX, Leica, Germany) with filter sets for DOX (excitation/emission: 488/520 nm). **All DOX diffusion profiles are measured by this method...**”

Comment #19: Please include the depth of injection for the intracranial murine tumor placement.

Our response: We thank the reviewer for the comment. The tumour was injected using a Hamilton syringe, which was positioned at AP -1.3 mm, ML +2.0 mm, and DV -1.0 mm.

Our modification: (Line 32, page 8: in revised Supplementary Information)

“...Total 3×10^6 cells were injected per mouse using a Hamilton syringe fitted with a 28-gauge needle, which was positioned by the stereotaxic device. The following coordinates with the stereotaxic guidance were used: AP -1.3 mm, ML +2.0 mm, and DV -1.0 mm....”

Comment #20: *Suppl Figure 16b seems to be mislabeled. 16c appears to be the magnified version not 16b.*

Our response: We thank the reviewer for the comment. We corrected the figure caption.

Our modification: (Supplementary Fig. 17b and c: in revised Supplementary Information)

“Supplementary Figure 17

Effect of DOX on survivin expression in U87-MG tumours at the indicated time-points. Survivin expression (a) without and (b) with heating ($\Delta T = 5^\circ\text{C}$). (c) Magnified image of (b).”

Comment #21: *Fig 18c is 5mm of human skull? Please clarify.*

Our response: We assumed the thickness of the human skull as 5 mm according to the SI references [4-6].

Our modification: (Supplementary Fig. 19b and c: in revised Supplementary Information)

“Supplementary Figure 19

The 3D FEM simulation results of contour plots depending on the different skull thickness. (a) No skull, (b) 0.15 mm thickness of skull (similar to mouse skull thickness), and (c) 5 mm thickness of skull (similar to human skull thickness).”

Comment #22: *In Suppl Fig 22d it appears that the wafer is sitting on top of the skull- Is the wafer placed intracranially or above the skull? Also please state how the mean tumor volumes calculated in (j).*

Our response: We thank the reviewer for the comment. The wafer was placed intracranially after making a burr hole. The mean tumour volumes were measured by integration of the tumour area by using the

sliced MR images with the resolution of 1mm/layer. We clarified the method in supplementary information.

Our modification: (Line 2, page 8: in revised Supplementary Information)

“...by MR imaging every week. The tumour area in each MR image slice was measured and multiplied by the layer thickness (1 mm). Then the summation of the volume from each slice was regarded as a total tumour volume...”

Reviewer #2:

Summary Comments: *In this manuscript, Lee and collaborators describe a new flexible, biodegradable, wireless, and activated by mild heat device for drug delivery to treat brain tumors. The authors show that the drug can be delivered to the tumor site without leakage. The device completely dissolved without debris left behind by being converted to biocompatible metabolites, within 10 weeks in the canine brain, a time that corresponds to the entire time of drug release. No inflammatory responses were detected. Finally, the authors provide data assessing the efficacy of the device in vivo. Experiments and results are well and clearly described. If efficacious in treating patients with brain tumors, it certainly be a great improvement compared to current therapeutic approaches.*

Our response: We are grateful for the reviewer's precise summary on our work. We hope that our research can contribute to the brain tumor treatment. We revised our manuscript according to the reviewer's comment.

Comment #1: *Because treatment with a single drug has proven to be ineffective, would the device be used to deliver combination therapies? Does the vehicle used for drug resuspension matters to keep the integrity of the device? The authors might want to address these two questions in the Discussion*

Our response: We thank the reviewer for pointing out this issue. The combination therapy using multiple anti-cancer agents has been proven to be more effective. We tested other agents such as temozolomide (TMZ) with the BEP (Supplementary Fig. 3d). Then, we additionally tested if the BEP can deliver both DOX and TMZ (Supplementary Fig. 3e). The BEP could work well as a drug delivery vehicle for this combination therapy. Although we loaded two kinds of drugs, the integrity of the device was well maintained. The glycerol, used as a plasticizer in the BEP fabrication process, did not break the integrity of the device.

Our modification: (Line 17, page 5: in revised main text)

“...Temozolomide¹¹³³ can also be used as an alternative drug (Supplementary Fig. 3d). The BEP can also deliver multiple anti-cancer agents (DOX and TMZ) for the combination therapy (Supplementary Fig. 3e). Although we loaded two kinds of drugs in the BEP, the integrity of the device was well maintained...”

(Supplementary Fig. 3e: in revised Supplementary Information)

(e) OST patch containing both 20 mg of DOX and 20 mg TMZ.

Comment #2: Results in Figure 2 h and I are difficult to see. Quantification of the immunohistochemistry would help.

Our response: We are grateful for the reviewer’s comment. We quantified the GFAP- and Iba-1-stained cells in the sham and BEP implantation group near the surgery site to study the foreign body response. 7-8 BALB/c nude mice that we used to study the BEP were used for the investigation over 4 time points (1 day, 2 weeks, 4 weeks, and 6 weeks from the implant surgery). The sham group exhibited the accumulation of astrocytes and microglia from 2 weeks, but the degree of accumulation was maintained up to 6 weeks, which suggests no significant increase of the immune response after 2 weeks (Fig. 2h and i). Also, the accumulation of astrocytes and microglia in the BEP implantation group is similar to the sham group. According to these results, the BEP did not induce significant foreign body responses up to 6 weeks. We modified the text accordingly.

Our modification: (Line 13, page 7: in revised main text)

“...In order to examine biocompatibility, the BEPs were implanted on the surface of the surgical cavity made by the brain surgery in BALB/c nude mice. The distribution of astrocytes and microglia near the surgical cavity was observed in both the sham (Fig. 2h and i, red) and BEP implantation (Fig. 2h and i, blue) group at various time points (1 day, 2 weeks, 4 weeks, and 6 weeks). No significant increase of the migrated astrocytes and microglia was observed after 2 weeks, and the differences of the migrated astrocytes and microglia between the sham group and the BEP group were not significant over all time periods. The results suggest that the BEP did not induce the significant immune response...”

(Fig. 2h and i: in revised main text)

Quantification of the immunohistochemistry using BALB/c nude mice at different time points for the sham (red) and BEP implantation (blue) group (n = 7-8 for each group and time): (h) for GFAP and (i) for Iba-1. (NS = not significant)

(Supplementary Fig. 8: in revised Supplementary Information)

Supplementary Figure 8

(a) Confocal fluorescence microscopy images of the GFAP expression in the tissue slices from the implantation site in BALB/c nude mice groups (top for sham and bottom for BEP group) at different time points (1 day, 2 weeks, 4 weeks, and 6 weeks). (b) Confocal fluorescence microscopy images of the Iba-1 expression in the tissue slices from the implantation site in BALB/c nude mice groups. Other conditions are same with (a).

(Line 8, page 15: in revised Supplementary Information)

“2.8. Histology.

Tissues were fixed in 10% formalin, incubated in graded ethanol, embedded in paraffin, and cut into 4-µm-thick sections. The histological analyses were performed with H&E (hematoxylin and eosin) staining for basic morphological evaluation and TUNEL (terminal deoxynucleotidyl transferase dUTP nick end labeling) staining for apoptotic cell detection. For Immunohistochemistry (IHC), all tissue sections were deparaffinized in xylene and hydrated by immersion in a graded ethanol series. Antigen retrieval was performed in a microwave by placing the sections in epitope retrieval solution (0.01 M citrate buffer, pH 6.0) for 20 min; endogenous peroxidase was inhibited by immersing the sections in 0.3% hydrogen peroxide for 10 min. The sections were then incubated with a primary mouse monoclonal antibody to MAC387 (ThermoFisher Scientific, USA) for inflammatory macrophages in a canine model or a rabbit monoclonal antibody to survivin (Cell Signaling, USA) in Dako REAL antibody diluent (Dako, USA) for a mouse model. The IHC staining of GFAP and Iba-1 for biocompatibility test was performed with the rabbit monoclonal anti-GFAP (Abcam, USA) and anti-Iba-1 (Wako Chemicals, USA) using the Dako Autostainer Link 48 system according to recommended protocols. Two region of interest (ROI) from each slide were acquired and all images were analyzed using Image J....”

Comment #3: The authors use the word actuation throughout the manuscript. Do the authors mean activation as in “mild-thermic activation” instead of “mild-thermic actuation”

Our response: We thank the reviewer for the comment. Both “activation” and “actuation” can be used

for the meaning to make something active. However, we think that they still have slightly different meanings. While “activation” makes something to become active, the “actuation” makes something to work in an intended way. Since we designed the wireless actuator to increase the surrounding tissue temperature and accelerate the drug diffusion through the cell membrane, we used the term of “mild-thermic actuation” for our application.

Reviewer #3:

Summary Comment: *In their manuscript “Flexible, Sticky, and Biodegradable Wireless Device for Drug Delivery to Brain Tumours” Lee, Cho, Cha and colleagues discuss the development of a bioresorbable drug-delivery patch thermally driven by eddy currents triggered within an integrated metal patch by an externally applied radiofrequency alternating magnetic field. The team evaluated their device in xenograft models of glioblastoma (GBM) in immunodeficient mice and dogs and corroborated the enhanced ability to deliver doxorubicin (DOX) to the tumor resection sites. The latter resulted in improved survival rates in the mouse GBM model. While ambitious, innovative, and extensive in its scope this study leaves a few technical issues unaddressed and could be further improved through additional analyses. My specific comments are detailed below.*

Our response: We are thankful for the reviewer’s valuable summary and insightful comments. We have revised our manuscript in a point-by-point manner according to the reviewer’s comments.

Comment #1: *The use of eddy current heating to facilitate the DOX delivery to the tumor resection site is an interesting concept. The technical description of the magnetic field conditions and the overall device design, however, is somewhat nebulous. The authors use position of the driving field coil with respect to the eddy heater as a key parameter of their design. This is meaningless unless precisely the same coil is applied for all future applications of this technology. The latter is unlikely given the fact that the field coil and the driving electronics are designed for induction welding and are not optimized for biomedical purposes. The authors should quantify the amplitude of the magnetic field rather than distance and calculate the resulting eddy currents in their heater.*

Our response: We are very grateful for the reviewer’s comment. We agree with the reviewer in that the heating, even at the same coil-to-heater distance, can vary depending on situations *in vivo*. Therefore, we added a plot (Supplementary Fig. 13c) that shows the relation between the magnetic field and the temperature increase of the heater, and related texts. The calculation of the eddy current and the related temperature increase are explained in *Comment #2*.

Our modification: (Line 1, page 9: in revised main text)

“...our instrument to set the temperature change. Under different conditions, the calibration curve that shows the temperature increase as a function of the magnetic field (Supplementary Fig. 13c) and/or the total eddy current (Supplementary Fig. 13d) can be used to optimize the amount of heating...”

(Line 10, page 5: in revised Supplementary Information)

“...following the external wireless mild-thermic actuation. The magnetic field generated from the transmission coil can be calculated as follows. The magnetic field intensity (B_1) generated from the coil is expressed as,

$$B_1 = \frac{\mu_0 I a^2}{(a^2 + h^2)^{1.5}}$$

where h is the distance from the center of the coil, a is the radius of the coil, I is the current of the coil, and μ_0 is the permeability of vacuum. If the coil has several turns, the total sum of the magnetic field (B) can be calculated as,

$$B = \sum_{i=1}^N \frac{\mu_0 I a^2}{(a^2 + h_i^2)^{1.5}}$$

where N is the turn number of the coil. For simplification, the distances from each turn of the coil to the heater, h_i , can be considered to be same as h . Then the equation becomes as follows,

$$B = N \frac{\mu_0 I a^2}{(a^2 + h^2)^{1.5}}$$

Using this equation, the relation between the magnetic field and the temperature increase can be plotted as shown in Supplementary Fig. 13c.”

(Supplementary Fig. 13c in revised Supplementary Information)

(c) Temperature increase by the wireless heater of various diameters as a function of the external magnetic field generated by the transmission coil.

Comment #2: They should also elaborate on the principles underlying the design of the eddy heater – what were the key optimization criteria? Would a field with a higher amplitude or frequency afford further miniaturization of the heater? The calibration curve for the temperature as a result of eddy current should be provided (T vs. I or T vs. H).

Our response: Theoretically, the amount of heat generation (W) is proportional to the integrated value of eddy currents in the heater (Equation 1 and 2), which is a function of the coil-to-heater distance (h), current of the transmission coil (I), number of the coil turn (N), coil radius (a), RF frequency (ω), heater thickness (t), and heater diameter (D) (Equation 3), assuming the device as a round-shaped magnesium plate. Since N , a , and ω are fixed by the instrument (magnetic field generator), the key parameters for the design are thickness (t) and diameter (D) of the wireless heater. The other parameters, coil-to-heater distance (h) and current in transmission coil (I), can also be optimized during the experiment. However, their variation is somehow limited; the coil-to-heater distance should be larger than 2-3 cm for the large animal application, and the coil current is limited by the instrument specification. Therefore, the most important parameters are heater thickness (t) and diameter (D), which are expressed as bold in equation 3.

$$J_{\phi}(\rho) = \frac{\sigma \omega B(\rho) \rho}{2} \dots\dots\dots (1)$$

$$W = \int \frac{J_{\phi}^2}{2\sigma} dv = \frac{\pi\sigma t \omega^2 B^2 D^4}{256} = \frac{N^2 \omega^2 \mu_0^2 I^2 a^4 D^4}{(a^2 + h^2)^3} \frac{\pi\sigma t}{256} \dots\dots\dots (2)$$

$$W = f(N, a, \omega, h, I, t, D) \dots\dots\dots (3)$$

Because of practical limitations of the physical vapor deposition (PVD) process, we fixed the heater thickness (3 μm) but, instead, changed the heater diameter. Another important design consideration of the wireless heater design is the hole array which is needed for the facile device fabrication (transfer-printing) process. The presence of holes slightly affects the amount of the heat generation (Supplementary Fig. 13a and b). Based on these design considerations, the current heater design could be optimized to generate sufficient heating (Fig. 3c) for the mild-thermic actuation.

Second, as the reviewer pointed out, the higher coil current (I) and/or frequency (ω) can decrease the size of the heater further. The theoretical analysis shows that heaters of the smaller diameter can generate the similar amount of heat under the larger current (I : Fig. 3c inset), and the simulation model shows that the similar amount of heat can be generated under a higher frequency (ω : Supplementary Fig. 13e and f).

We added a plot of the temperature increase as a function of the magnetic field (Supplementary Fig. 13c) and total eddy current (Supplementary Fig. 13d) using above equations. We also added detailed explanations in the main text and Supplementary Information accordingly.

Our modification: (Line 19, page 8: in revised main text)

“...resulting in the increased temperature of the BEP and surrounding brain tissues. **Key parameters of the heater design are the diameter and thickness of the round-shaped heater while the transmission coil current and coil-to-heater distance can be varied to optimize the heat generation. The hole array in the wireless heater is helpful for facile fabrication (transfer-printing) of the device but slightly affects the heat generation (Supplementary Fig. 13a and b). The temperature change under the various coil-to-heater distances in the heater (Fig. 3c) of different diameters and under the various coil-currents in 12 mm diameter heater (Fig. 3c inset) were measured using our instrument to set the temperature change. Under different conditions, the calibration curve that shows the temperature increase as a function of the magnetic field (Supplementary Fig. 13c) and/or the total eddy current (Supplementary Fig. 13d) can be used to optimize the amount of heating. Also, the dimension of the heaters (thickness and diameter) can be decreased or the coil-to-heater distance can be increased further by controlling the instrumental parameters such as the RF frequency (Supplementary 13e and f). More details are described in Supplementary Text 1.2...**”

(Line 28, page 5: in revised Supplementary Information)

“...We assumed the magnetic field to be constant over the small area of the heater with the diameter of 12 mm at the distance of 3 cm from the coil (Quasi-static assumption). Internal layers between the transmission coil and the heater are modeled as a nonmagnetic material. Then the eddy current is calculated as,

$$J_{\phi}(\rho) = \frac{\sigma\omega B(\rho)\rho}{2}$$

where J is the eddy current, ρ is the distance from the center of the heater, σ is the electrical conductivity of the Magnesium, ω is the frequency of the coil. Although the eddy current of the BEP with holes cannot be solved analytically, a simple assumption, the round heater model without holes, can be introduced for identifying the relationship between variables. Then, the Joule heating amount is as follows,

$$W = \int \frac{J_{\phi}^2}{2\sigma} dv = \frac{\pi\sigma t \omega^2 B^2 D^4}{256} = \frac{N^2 \omega^2 \mu_0^2 I^2 a^4 D^4 \pi \sigma t}{(a^2 + h^2)^3 256}$$

where W is the amount of the heat generation that is determined by the coil-to-heater distance (h), current of the transmission coil (I), number of the coil turn (N), coil radius (a), RF frequency (ω), heater thickness (t), and heater diameter (D). According to the above equation, the heat generation is proportional to the square of the coil current and the square of the RF frequency. Since N , a , and ω are fixed by the instrument (magnetic field generator), the key design parameters are the thickness (t) and diameter (D) of the wireless heater. The other parameters, coil-to-heater distance (h) and current in transmission coil (I), can also be optimized during the experiment.

For the analysis of the round heater with the hole array, the HFSS simulation is used. The heaters of three different diameters show similar amount of heat generation under the three different frequencies, which shows the potential for further minimization of the wireless heater (Supplementary Fig. 13e and f).”

(Supplementary Fig. 13 in revised Supplementary Information)

Supplementary Figure 13

Theoretical analysis and HFSS simulation of the wireless heater under different mild-thermic actuation conditions. 2-D Contour plot of the eddy current distribution of the BEP (a) with and (b) without the hole array under the optimized frequency. (c) Temperature increase by the wireless heater of various diameters as a function of the external magnetic field generated by the transmission coil. (d) Temperature increase by the wireless heater of 12 mm diameter as a function of the total eddy current. (e) 1-D eddy current

from the center to the edge of the BEP with different frequencies and heater sizes. (f) Heat generation of the BEP with different frequencies and different heater sizes.

Comment #3: *The authors emphasize the degradable nature of their patch. Simultaneously they design the patch to be unidirectionally functional, releasing drugs toward the brain while avoiding excessive leakage into the epidural or subdural space. I am very perplexed by this concept. First, the tumor cavity is unlikely to have a uniform shape enabling precise placement of the patch in a manner that only permits release into the brain. Second, as the device degrades, the drug is expected to leak essentially everywhere in the cavity. The authors should show profiles of the drug leakage overlaid with device degradation – immunohistochemistry with the device still in place over an entire operation window would be most enlightening.*

Our response: We thank the reviewer for valuable comments. As the reviewer pointed out, the perfect unidirectional diffusion cannot be achieved due to incomplete coverage of the tumour cavity by the BEP particularly during its biodegradation. What we intended was to reduce/minimize the drug release to the CSF. Therefore, we modified the text to clarify this issue. We additionally carried out the histology analysis to confirm the suppressed drug leakage. We observed the fluorescence of the drug at 3 different locations (Supplementary Fig. 6a) to visualize the drug leakage to the nearby tissue. Compared to the site 1 (implanted position), the site 2 (nearby position) and site 3 (distant position) exhibited much lower fluorescence, which means that the concentration of the drug is much lower. We added the data and the related texts accordingly.

Our modification: (Line 21, page 3: in revised main text)

“...its hydrophilic/hydrophobic bifacial design allow conformal adhesion to the target brain tissue^{24,25} and enable **local** and sustained drug delivery...”

(Line 18, page 6: in revised main text)

“...suppresses the drug release to the PBS solution dramatically (Fig. 2d red). **The selective coating of PLA only on the top surface of the device where the device is exposed to CSF can decrease the unintended drug leakage to other regions (Supplementary Fig. 5 and 6; *in vitro* and *in vivo*, respectively), although perfect prevention of the drug diffusion to the CSF cannot be achieved due to biodegradation.** Meanwhile, the bottom hydrophilic OST substrate enhances adhesion of the BEP to the brain surface and...”

(Line 19, page 4: in revised main text)

“...These conformal and strong adhesion improves the **efficiency** of **the** drug delivery. Meanwhile, the hydrophobic PLA top encapsulation **reduces** undesirable drug delivery to CSF...”

(Supplementary Fig. 6: in revised Supplementary Information)

Supplementary Figure 6

Imaging of DOX at three different sites (1: center of the implantation site (the parietal lobe), 2: near the implantation site (the parietal lobe), 3: far from the implantation site (the occipital lobe)) in the canine brain at various time points. (a) Gross image of the canine brain at day 1 after the BEP implantation. Fluorescence imaging of DOX at three different sites in the canine brain at (b) 1 day, (c) 2 weeks, (d) 4 weeks, (e) 6 weeks after the BEP implantation.

Comment #4: The authors perform their adhesion tests on the bovine muscle tissue. But the intended application is the surface of the brain. The two types of tissue have rather different mechanical properties and it is unclear why the bovine muscle was used as a testbed. Adhesion tests on the actual mouse or canine brain would be more representative.

Our response: We thank the reviewer for raising this issue. The mechanical strength of the brain tissue [1] is much lower than the adhesion force between the patch and brain. Therefore, the brain tissue becomes mechanically torn before the detachment of the device from the tissue surface in the adhesion test [2]. Although the bovine muscle is different from the brain tissue, its mechanical strength is stronger than the brain tissue, and thus the adhesion test could be proceeded, which showed the enhanced adhesion of the patch to the brain tissue indirectly. We added this point in the revised manuscript.

[1] Budday, S. *et al.* Mechanical properties of gray and white matter brain tissue by indentation. *J. Mech. Behav. Biomed.* **46**, 318-330, (2015).

[2] Yuk, H., Zhang, T., Lin, S., Parada, G. A. & Zhao, X. Tough bonding of hydrogels to diverse non-porous surfaces. *Nat. Mater.* **15**, 190, (2015).

Our modification: (Line 21, page 5: in revised main text)

“...Since the adhesion force between the OST film and the brain tissue is stronger than the mechanical strength of the brain tissue^{34, 35}, the brain tissue is mechanically torn before the detachment of the film from the brain surface. Therefore, the effect of oxidization of starch on the adhesion strength was indirectly tested on the bovine muscle instead of the brain tissue, since the bovine muscle has higher mechanical strength than the brain tissue. The shear adhesion test (Fig. 2b inset) of OST exhibited its strong adhesion to the muscle tissues (Fig. 1a right) than that of the unmodified starch film (non-OST case, 0% in Fig. 2b) due to imine conjugation (more oxidised units)...”

(Reference: in revised main text)

34. Budday, S. *et al.* Mechanical properties of gray and white matter brain tissue by indentation. *J. Mech. Behav. Biomed. Mater.* **46**, 318-330 (2015).
35. Yuk, H., Zhang, T., Lin, S., Parada, G.A. & Zhao, X. Tough bonding of hydrogels to diverse non-porous surfaces. *Nat. Mater.* **15**, 190 (2015).

Comment #5: Immunohistochemistry analyses of the foreign body response are somewhat qualitative. The data should be shown for multiple animals (typically $n=5-8$) for several time points within the lifespan of the device. These data should be collected in the brain of the same mouse models used for GBM evaluation experiments.

Our response: We are grateful for the reviewer's comment. We quantified the GFAP- and Iba-1-stained cells in the sham and BEP implantation group near the surgery site to study the foreign body response. 7-8 BALB/c nude mice that we used to study the BEP were used for the investigation over 4 time points (1 day, 2 weeks, 4 weeks, and 6 weeks from the implant surgery). The sham group exhibited the accumulation of astrocytes and microglia from 2 weeks, but the degree of accumulation was maintained up to 6 weeks, which suggests no significant increase of the immune response after 2 weeks (Fig. 2h and i). Also, the accumulation of astrocytes and microglia in the BEP implantation group is similar to the sham group. According to these results, the BEP did not induce significant foreign body responses up to 6 weeks. We modified the text accordingly.

Our modification: (Line 13, page 7: in revised main text)

“...In order to examine biocompatibility, the BEPs were implanted on the surface of the surgical cavity made by the brain surgery in BALB/c nude mice. The distribution of astrocytes and microglia near the surgical cavity was observed in both the sham (Fig. 2h and i, red) and BEP implantation (Fig. 2h and i, blue) group at various time points (1 day, 2 weeks, 4 weeks, and 6 weeks). No significant increase of the migrated astrocytes and microglia was observed after 2 weeks, and the differences of the migrated astrocytes and microglia between the sham group and the BEP group were not significant over all time periods. The results suggest that the BEP did not induce the significant immune response...”

(Fig. 2h and i: in revised main text)

Quantification of the immunohistochemistry using BALB/c nude mice at different time points for the sham (red) and BEP implantation (blue) group ($n = 7-8$ for each group and time): (h) for GFAP and (i) for Iba-1. (NS = not significant)

(Supplementary Fig. 8: in revised Supplementary Information)

Supplementary Figure 8

(a) Confocal fluorescence microscopy images of the GFAP expression in the tissue slices from the implantation site in BALB/c nude mice groups (top for sham and bottom for BEP group) at different time points (1 day, 2 weeks, 4 weeks, and 6 weeks). (b) Confocal fluorescence microscopy images of the Iba-1 expression in the tissue slices from the implantation site in BALB/c nude mice groups. Other conditions are same with (a).

(Line 8, page 15: in revised Supplementary Information)

“2.8. Histology.

Tissues were fixed in 10% formalin, incubated in graded ethanol, embedded in paraffin, and cut into 4- μ m-thick sections. The histological analyses were performed with H&E (hematoxylin and eosin) staining for basic morphological evaluation and TUNEL (terminal deoxynucleotidyl transferase dUTP nick end labeling) staining for apoptotic cell detection. For Immunohistochemistry (IHC), all tissue sections were deparaffinized in xylene and hydrated by immersion in a graded ethanol series. Antigen retrieval was performed in a microwave by placing the sections in epitope retrieval solution (0.01 M citrate buffer, pH 6.0) for 20 min; endogenous peroxidase was inhibited by immersing the sections in 0.3% hydrogen peroxide for 10 min. The sections were then incubated with a primary mouse monoclonal antibody to MAC387 (ThermoFisher Scientific, USA) for inflammatory macrophages in a canine model or a rabbit monoclonal antibody to survivin (Cell Signaling, USA) in Dako REAL antibody diluent (Dako, USA) for a mouse model. The IHC staining of GFAP and Iba-1 for biocompatibility test was performed with the rabbit monoclonal anti-GFAP (Abcam, USA) and anti-Iba-1 (Wako Chemicals, USA) using the Dako Autostainer Link 48 system according to recommended protocols. Two region of interest (ROI) from each slide were acquired and all images were analyzed using Image J....”

Comment #6-1: *The authors discuss using EEGs as a means to confirm the “normal” state of the brain tissue following implantation of their patches. This entire part of the manuscript is rather confusing. First, looking at the electrode positioning within the brain, it appears that the authors used penetrating electrodes placed within the brain tissue following craniotomy. A typical EEG is performed using electrodes placed on the surface of the skull.*

Our response: We appreciate the reviewer’s comment. In case of the EEG measurement for the mouse model, EEG can be measured by using the penetrating electrode [1] or the electrode attached on dura [2]. The former method is better in detection of partial seizures although the latter can be used for detection of a big cramp such as epilepsy. The EEG study in this work is to see the influence of the BEP on the electrophysiological activity, and therefore we chose to use the penetrating electrode to detect any potential partial seizures [1]. But to prevent confusion from the conventional surface EEG measurement, we changed the terminology from EEG to iEEG (intracranial EEG) in our revised manuscript.

[1] Wu, C., Wais, M., Zahid, T., Wan, Q. & Zhang, L. An improved screw-free method for electrode implantation and intracranial electroencephalographic recordings in mice. *Behavior Res. Meth.* **41**, 736-741, (2009).

[2] Weiergräber, M., Henry, M., Hescheler, J., Smyth, N. & Schneider, T. Electrographic and deep intracerebral EEG recording in mice using a telemetry system. *Brain Res. Protoc.* **14**, 154-164 (2005).

Our modification: (Line 23, page 7: in revised main text)

“...The **intracranial electroencephalogram (iEEG)** was also monitored and analyzed (Supplementary Fig. 9c)...”

(Line 29, page 16: in revised Supplementary Information)

“...Electroencephalographic signals in the mice were continuously recorded for 2 weeks. **The epileptologist interpreted the intracranial electroencephalogram (iEEG) based on the occurrence of rhythmic burst discharges...**”

(Supplementary Fig. 9c: in revised Supplementary Information)

“(c) **Intracranial electroencephalograms (iEEGs)** of the sham operating group and the BEP implantation group at different time points.”

Comment #6-2: *Second, it is unclear how the data was recorded and filtered. Supplementary Figure 9c does not appear to have any scale bars on either axis. It not even clear what's on the axes.*

Our response: We are thankful for the reviewer’s comment. The tungsten electrodes were positioned in (AP, ML, DV= -1.8 mm, 2.1 mm, 0.8–1.0 mm) from the bregma with grounding over the cerebellum. The electrical activities were recorded after amplification ($\times 1200$), band-pass-filter from 0.1 to 70 Hz, and digitization at a 400-Hz sampling rate (AS 40) with a digital electroencephalography system (Comet XL; Astro-Med, Inc., Warwick, RI, USA). Electrophysiological data were analyzed offline using PSG Twin 4.2 (Astro-Med, Inc.). Electroencephalographic signals in the mice were continuously recorded for 2 weeks. The detailed description regarding this procedure can be found in Supplementary Text 2.1. We added the voltage/time scale bars to Supplementary Fig. 9c.

Our modification: (Supplementary Figure. 9c: in revised Supplementary Information)

(c) Intracranial electroencephalograms (iEEGs) of the sham operating group and the BEP implantation group at different time points.

Comment #6-3: Third, what features did the authors use to confirm that the electrophysiological activity is “normal”. What constitutes normal for this particular electrode placement? What would be a signature of an abnormality?

Our response: We thank for the reviewer’s comment. The clinical epileptologist interpreted the iEEG signal to check any occurrence of the rhythmic burst discharges. EEG without any significant burst discharges was regarded as a normal status. We modified the manuscript to explain this point.

Our modification: (Line 29, page 16: in revised Supplementary Information)

“...Electroencephalographic signals in the mice were continuously recorded for 2 weeks. The epileptologist interpreted the intracranial electroencephalogram (iEEG) signal to check any occurrence of the rhythmic burst discharges...”

Comment #7: It is unclear what motivated the choice of the rotarod motor performance as a metric to corroborate the device biocompatibility. Was the device implanted above the motor cortex?

Our response to comment #7: We appreciate the reviewer’s comment. Although the BEP is implanted next to the motor cortex, we hypothesized that the BEP might affect the motor cortex, if there are negative tissue responses such as edema. We modified the text for clarification.

Our modification: (Line 32, page 10: in revised Supplementary Information)

“...Rotarod performance of mice was assessed as previously described¹⁰ for verifying the indirect influence of the BEP implanted adjacent to motor cortex. The accelerating Rota-Rod (San Diego Instruments, San Diego, CA, USA) was used to linearly increase the speed from 4 to 40 rpm over 3 min...”

Other modifications:

Two authors who contributed to the revision, one for the wireless power transfer and the other for clinical neurology, were newly added to the author list.

Modification to the author list:

Jongha Lee^{1,2,†}, Hye Rim Cho^{1,3,†}, Gi Doo Cha^{1,2,†}, Hyunseon Seo^{1,2,4}, Seunghyun Lee³, Chul-Kee Park⁵, Jin Wook Kim⁵, Shutao Qiao⁶, Liu Wang⁶, Dayoung Kang^{1,2}, Taegyung Kang^{1,2}, Tomotsugu Ichikawa⁷, Jonghoon Kim^{1,2}, Hakyong Lee^{1,2}, Woongchan Lee^{1,2}, Sanghoek Kim⁸, Soon-Tae Lee⁹, Nanshu Lu⁶, Taeghwan Hyeon^{1,2}, Seung Hong Choi^{1,3,*}, Dae-Hyeong Kim^{1,2}

¹Center for Nanoparticle Research, Institute for Basic Science (IBS), Seoul 08826, Republic of Korea.

²School of Chemical and Biological Engineering, Institute of Chemical Processes, Seoul National University, Seoul 08826, Republic of Korea.

³Department of Radiology, Seoul National University College of Medicine, Seoul 03080, Republic of Korea.

⁴Center for Biomaterials, Korea Institute of Science and Technology, Seoul 02792, Republic of Korea.

⁵Department of Neurosurgery, Seoul National University College of Medicine, Seoul 03080, Republic of Korea.

⁶Center for Mechanics of Solids, Structures and Materials, Department of Aerospace Engineering and Engineering Mechanics, University of Texas at Austin, TX 78712, USA.

⁷Department of Neurological surgery, Okayama University Graduate School of Medicine, Dentistry, and Pharmaceutical Sciences, Okayama 700-8558, Japan.

⁸ Department of Electronics and Radio Engineering, Kyung Hee University, Gyeonggi 17194, Republic of Korea.

⁹Department of Neurology, Seoul National University College of Medicine, Seoul 03080, Republic of Korea.

[†]J. Lee, H. R. Cho, and G. D. Cha contributed equally to this work.

*To whom correspondences should be addressed.

E-mail: dkim98@snu.ac.kr and verocay@snuh.org.

Errors in experimental conditions were corrected.

(Line 22, page 7: in revised Supplementary Information)

“Implantation of human glioblastoma cells into mice. Thirty-three 6-week-old female BALB/c mice weighing approximately 20–25 g were used in this study...”

→

“Implantation of human glioblastoma cells into mice. Thirty-three 6-week-old female BALB/c **nude** mice weighing approximately 20–25 g were used in this study...”

REVIEWERS' COMMENTS:

Reviewer #1 (Remarks to the Author):

brief, I believe this is an extremely exciting, breakthrough very high impact discovery that has in a highly original manner solved many of the existing problems to complex controlled drug delivery to the brain. It is extremely important to brain tumor researchers but even more so to all neuroscientists as it shows a novel way of externally controlling drug delivery to the brain.

I strongly support rapid publication as a high priority high impact article on highly original and important work with wide implications in neuroscience research and brain tumor therapy.

Reviewer #2 (Remarks to the Author):

The authors have addressed my criticisms adequately. Some editing is still required before publication.

Reviewer #3 (Remarks to the Author):

The authors have done a thorough job addressing my comments, and the manuscript has been strengthened substantially. I remain, however, confused with the electrophysiological and behavioral data, which do not actually add to this otherwise excellent manuscript.

It appears that the “normalcy” of EEGs was based on the absence of seizures. However, the latter are an extreme condition and many subtle electrophysiological differences may take place that do not constitute a seizure yet could be classified as abnormal. Hence, I suggest that the authors explicitly state that they were controlling for seizures and motivate why those are relevant in the context of their work. Similarly, for their rotarod experiments, they should motivate why they were concerned about the motor abnormalities given the fact that the procedure was performed close to but not in the motor cortex.

I consider these changes to be minor requiring merely changes to the writing and motivation within the text.

Reviewer #1

Summary Comments: brief, I believe this is an extremely exciting, breakthrough very high impact discovery that has in a highly original manner solved many of the existing problems to complex controlled drug delivery to the brain. It is extremely important to brain tumor researchers but even more so to all neuroscientists as it shows a novel way of externally controlling drug delivery to the brain. I strongly support rapid publication as a high priority high impact article on highly original and important work with wide implications in neuroscience research and brain tumor therapy.

Our response: We are grateful for the reviewer's supportive and positive comments on our manuscript. As the reviewer pointed out, we believe that our work can contribute to the fields of neuroscience and brain tumour research.

Reviewer #2

Summary Comments: The authors have addressed my criticisms adequately. Some editing is still required before publication.

Our response: We thank the reviewer for the insightful comments that have improved the quality of our manuscript significantly. We revised our manuscript to provide a higher-quality research article via additional editing.

Reviewer #3

Summary Comments: The authors have done a thorough job addressing my comments, and the manuscript has been strengthened substantially.

Our response: We are pleased for the reviewer's comment on our manuscript.

Comment #1: I remain, however, confused with the electrophysiological and behavioral data, which do not actually add to this otherwise excellent manuscript. It appears that the "normalcy" of EEGs was based on the absence of seizures. However, the latter are an extreme condition and many subtle electrophysiological differences may take place that do not constitute a seizure yet could be classified as abnormal. Hence, I suggest that the authors explicitly state that they were controlling for seizures and motivate why those are relevant in the context of their work. Similarly, for their rotarod experiments, they should motivate why they were concerned about the motor abnormalities given the fact that the procedure was performed close to but not in the motor cortex.

I consider these changes to be minor requiring merely changes to the writing and motivation within the text.

Our response: We appreciate the reviewer's valuable comment. We added the related contexts in our Supplementary Methods 2.10.

Our modification: (Line 22, page 14: in revised Supplementary Information)

"In vivo electroencephalography surgery were performed as previously described^{8,9}. This is for detecting whether a severe degree of seizure may occur due to the implanted BEP."

(Line 6, page 15: in revised Supplementary Information)

“Rotarod performance of mice was assessed as previously described¹⁰ for verifying the indirect influence of the BEP implanted adjacent to the motor cortex, **such as potential physical and/or chemical damages caused by implantation surgery, intracranial pressure change, and anti-tumour drug release.**”

Again, we appreciate your time and effort.